# ARP-T1-associated Bazex–Dupré–Christol syndrome is an inherited basal cell cancer with ciliary defects characteristic of ciliopathies

Hyun-Sook Park[1,7], Eirini Papanastasi[1,7], Gabriela Blanchard[1], Elena Chiticariu[1], Daniel Bachmann[1], Markus Plomann [2], Fanny Morice-Picard[3], Pierre Vabres [4], Asma Smahi [5,6], Marcel Huber[1], Christine Pich [1] & Daniel Hohl[1✉]

Actin-Related Protein-Testis1 (ARP-T1)/*ACTRT1* gene mutations cause the Bazex-Dupré-Christol Syndrome (BDCS) characterized by follicular atrophoderma, hypotrichosis, and basal cell cancer. Here, we report an ARP-T1 interactome (PXD016557) that includes proteins involved in ciliogenesis, endosomal recycling, and septin ring formation. In agreement, ARP-T1 localizes to the midbody during cytokinesis and the basal body of primary cilia in interphase. Tissue samples from ARP-T1-associated BDCS patients have reduced ciliary length. The severity of the shortened cilia significantly correlates with the ARP-T1 levels, which was further validated by ACTRT1 knockdown in culture cells. Thus, we propose that ARP-T1 participates in the regulation of cilia length and that ARP-T1-associated BDCS is a case of skin cancer with ciliopathy characteristics.

[1] Department of Dermatology, CHUV-FBM UNIL, Hôpital de Beaumont, Lausanne, Switzerland. [2] Center for Biochemistry, University of Cologne, Cologne, Germany. [3] Department of Dermatology, CHU, Hôpital Pellegrin, Bordeaux, France. [4] Department of Dermatology, CHU, Hôpital du Bocage, Dijon, France. [5] Paris Descartes University, Sorbonne Paris Cité, Paris, France. [6] IMAGINE Institute INSERM UMR 1163, Paris, France. [7] These authors contributed equally: Hyun-Sook Park, Eirini Papanastasi. ✉email: daniel.hohl@chuv.ch

Basal cell cancer (BCC) of the skin is the most frequent human cancer. Bazex–Dupré–Christol syndrome (BDCS) is an X-linked dominantly inherited syndrome form of BCC, without male-to-male transmission, affecting primarily the hair follicle, resulting in the triad of hypotrichosis, follicular atrophoderma and BCC[1–3]. Hypotrichosis and follicular atrophoderma develop around birth and BCC early in adulthood. BDCS less frequently presents with milia, hypohidrosis, facial pigmentation and trichoepithelioma, and thus combines developmental cutaneous anomalies with tumor predisposition, i.e. an ectodermal dysplasia with a hereditary tumor syndrome[4,5]. Unlike most cases of BCC, which are sporadic, BDCS is an inherited syndromic form of BCC. The insertion mutation *ACTRT1 547_548InsA* creates a shift in the reading frame that results in a non-functional truncated protein of 198 amino acids. ARP-T1 was also found depleted in families with germline mutations of non-coding sequences surrounding *ACTRT1* postulated to belong to enhancers transcribed as non-coding RNAs. As these mutations segregate with the disease, ARP-T1 can be considered a tumor suppressor in BDCS[6]. ARP-T1 was first isolated as a major acidic component of the cytoskeletal calyx from the head of bull spermatozoa. ARP-T1 is expressed specifically late in spermatid differentiation in testis, where it locates to the postacrosomal region and the centriole[7], establishing a link to the primary cilium of epidermal cells implicated in BCC development.

Cilia are microtubule-based organelles that are either motile, such as sperm flagella, or nonmotile such as primary cilium that acts as sensory antenna, receiving signals from other cells nearby. The sensory capacity of the primary cilium is founded on the spatio-temporal localization of diverse receptor complexes such as PTCH, RTKs, TGFβR, NOTCH receptors, GPCRs including SMO, ion channels, and extracellular matrix receptors along the cilium[8]. Ciliary communication is often compromised in cancer[9], and faulty Hedgehog (HH) signaling implicated prototypically in medulloblastoma and BCC. Indeed, BCC cells are frequently ciliated, and activated HH signaling within primary cilia is a key driver in BCC pathogenesis[10]. BCCs generally show abnormal activation of the HH signaling pathway, ascribing its constitutive activation as a prerequisite for the tumor development[11]. HH signaling in its "off" state is characterized by PTCH1 repressing SMO activity. Primary cilia have a dual role and are required for or may repress tumor formation[12,13], depending on the level of the molecular interaction[9].

Primary cilium assembly is a complex process that involves the attachment of the mother centriole with its appendages to the plasma membrane in $G_1$ or upon cell cycle exit: the basal body, which nucleates the primary cilium with apical CEP164[14], a component of distal appendages, EHD1/EHD3 and SNAP29[15] to initiate primary vesicle formation and ciliogenesis. Assembly of the primary cilium includes recruitment of periciliary membranes, transition zone components, and a machinery of intraflagellar transport proteins to form the ciliary vesicle and to set the base for formation of the microtubular ciliary axoneme[15,16]. The primary cilium is separated from the cytosol by the basal body and anchored by transition fibers/distal appendages in the ciliary pocket or periciliary plasma membrane (PM)[17]. Transition fibers and transition zone with its Y-shaped linkers serve as a ciliary gate for the cilio-cytoplasmic transport sharing some functional similarities with the nuclear pore complex[17,18]. Polarized membrane trafficking to the cilium and the vesicular transport machinery, which targets Golgi derived vesicles and apical recycling endosomes containing essential cargo, such as GPCRs, to the periciliary PM or the ciliary pocket depends on small GTPases, notably RAB8 and RAB11, Rabin 8 and the BBSome, a stable complex of seven proteins implicated in the ciliopathy of Bardet-Biedl syndrome (BBS)[8].

Here, we show that ARP-T1 localizes to the basal body, interacts with several components of the ciliary machinery and contributes to cilium extension. Mutations in *ACTRT1* or its enhancer elements, as found in the tumor samples of BDCS patients, as well as ACTRT1 knockdown give rise to the abnormally shortened cilia, and this may be caused by the displacement of septin 2. Ciliopathies encompass most human organ systems in eyes, nose, ears, organ placement, energy homeostasis, infertility, skeleton, reproductive system, brain, hydrocephalus, heart, chronic respiratory problems, kidney, and liver[19]. We report for the first time the presence of cilia defects in ARP-T1-associated BDCS epidermal development and cancer, and propose that this pathology should be considered a ciliopathy.

## Results

**ARP-T1 is expressed in differentiated human keratinocytes and human epithelial cells**. We examined the expression of ARP-T1 in NHEK (Fig. 1a, b) and HaCaT (Fig. 1c, d) cultured under differentiating conditions with high calcium up to 7 days. The expression of ARP-T1 increased in a terminal differentiation dependent manner as indicated by the expression of keratin 10 (K10) in differentiating keratinocytes at both mRNA (Fig. 1a, c) and protein levels (Fig. 1b, d). To investigate whether ARP-T1 expression was only induced in keratinocytes or whether it was related to differentiation, we also analyzed its expression in two differentiating retinal-pigmented epithelial (RPE) cells, ARPE-19 (Fig. 1e, f) and hTERT-RPE1 (Fig. 1g, h). These cells were differentiated under serum starvation for 35 days or 48 h, respectively. During RPE differentiation, ciliogenesis is known to increase[14,20,21] and IFT88 is a well-characterized protein involved in RPE ciliogenesis[22,23]. We assessed RPE differentiation monitoring with the expression of the ciliary protein IFT88 (Fig. 1f, h), and with the increased number of ciliated cells (Supplementary Fig. 1a). Similar to keratinocytes, differentiated RPE cells exhibited an increased ARP-T1 expression after differentiation, at both mRNA (Fig. 1e, g) and protein levels (Fig. 1f, h, quantified in Supplementary Fig. 1b, c).

**ACTRT1 mRNA is regulated by non-canonical hedgehog pathway and protein kinase C**. ARP-T1 was previously reported to inhibit GLI1 expression and be involved in HH signaling[6]. We treated HH responsive hTERT-RPE1 cells with Smoothened (SMO) agonist (SAG), which binds to SMO and activates the HH pathway, and measured the expression of ACTRT1. GLI1 and PTCH1 mRNA expression was used as controls of HH pathway activation. GLI1 and PTCH1 mRNA responded well upon SAG treatment in proliferating and serum-starved cells (Fig. 2a, b, respectively). The expression of ACTRT1 mRNA and ARP-T1 protein was increased by SAG treatment in proliferating (Fig. 2a) and differentiating (Fig. 2b) conditions. We used additional SMO activator, purmorphamine, to confirm the results with SAG treatment, under differentiating condition. In this case, we also used vismodegib (SMO inhibitor used for the treatment of BCC) and both purmorphamine and vismodegib, to understand better the regulation of ARP-T1 by the HH pathway. ACTRT1, GLI1, and PTCH1 mRNA increased about two fold upon purmorphamine treatment similar to the treatment with SAG (Fig. 2c–e). GLI1 and PTCH1 expression decreased upon both purmorphamine and vismodegib treatment as expected (Fig. 2d, e), but not the expression of ACTRT1 indicating that ACTRT1 mRNA expression is regulated by a non-canonical HH pathway, although ARP-T1 protein expression was similar to the basal level under this condition (Fig. 2c).

A cascade of phosphorylation is known to be involved in HH signaling pathways, e.g. PKC, GSK-3β, S6K, PKA, PI3K, aPKC-ι/λ[8].

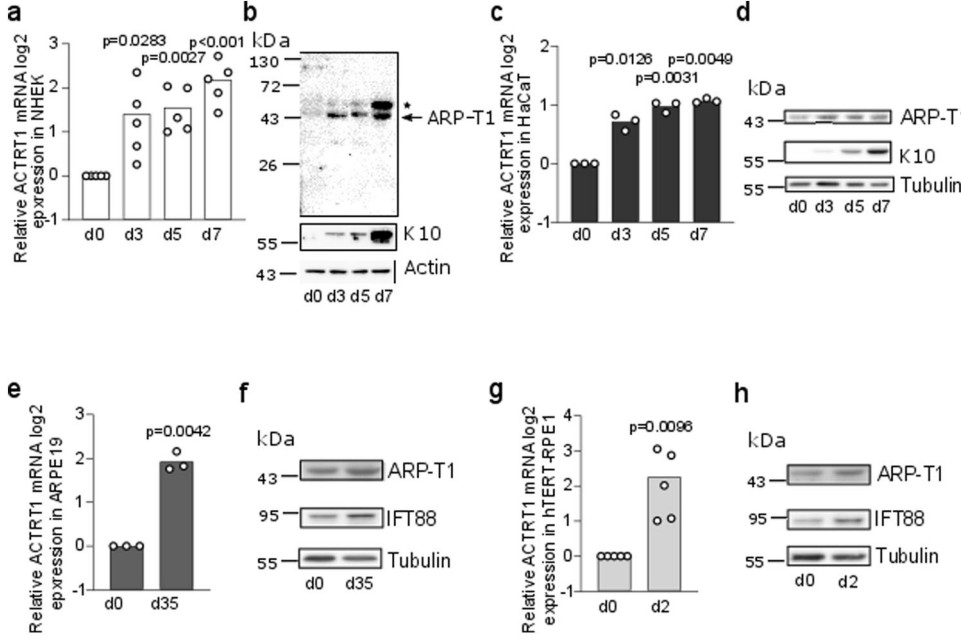

**Fig. 1 ARP-T1 is expressed during epidermal and epithelial differentiation. a**, **c**, **e**, **g** mRNA expression of ACTRT1 during differentiation of keratinocytes, NHEK (**a** $N = 5$) and HaCaT (**c** $N = 3$), and epithelial cells, ARPE19 (**e** $N = 3$) and hTERT-RPE1 (**g** $N = 5$). Data are presented as means of the fold change compared to the value of undifferentiated samples. Each open circle represents one independent experiment. **b**, **d**, **f**, **h** Representative images of ARP-T1 expression during differentiation of keratinocytes, NHEK (**b**) and HaCaT (**d**), and epithelial cells, ARPE19 (**f**) and hTERT-RPE1 (**h**). ARP-T1 was detected using guinea pig anti-ARP-T1 antisera (**b**) or mouse antibody (**d**, **f**, **h**). * indicates polymers of ARP-T1 confirmed by mass spectrometry analysis. Keratin 10 and IFT88 were used as markers of cell differentiation in keratinocytes and epithelial cells, respectively, actin and tubulin as loading controls.

To investigate the HH pathway in which ARP-T1 plays a role in BCC onset, differentiated NHEK cells, in which ARP-T1 is highly expressed, were treated with different kinase inhibitors. The cells treated with GF109203X known as PKC and PKA inhibitor showed significantly decreased expression of ACTRT1 mRNA, while cells treated with rapamycin (mTOR/S6K inhibitor), LY294002 (PI3K inhibitor), and AG1478 (epidermal growth factor tyrosine kinase inhibitor) showed increased expression of ACTRT1 mRNA (Fig. 2f). To further narrow down the findings we treated NHEK and HaCaT cells with Gö6976 (PKC alpha/beta inhibitor), rottlerin (PKC delta inhibitor), and ZIP 2549 (atypical PKC inhibitor). We found that rottlerin inhibits the expression of ACTRT1 mRNA in both NHEK and HaCaT cells, indicating that ACTRT1 mRNA expression is regulated by PKC delta (Fig. 2g, h). Two different programs (PhosphositePlus and Swiss Institute of Bioinformatics) predicted several putative phosphorylation sites in ARP-T1 WT and the mutant protein (from ACTRT1 547_548insA mutation) (Fig. 2i). One of the predicted phosphorylation sites in C-terminus at S352 position showed similar consensus phosphorylation site specificity with PKC (www.kinexus.ca). We transduced ACTRT1 (WT), ACTRT1 547_548insA (M), and vector control (V) tagged with DDK (same sequence as FLAG) into differentiated NHEK and immunoprecipitated using anti-FLAG M2 affinity gel. We used the Phospho-(Ser) PKC substrate antiserum, which recognizes proteins only when phosphorylated at serine residues surrounded by arginine or lysine at the −2 and +2 positions and a hydrophobic residue at the +1 position (ARP-T1 S352: MSS*FKQ). This antiserum recognized ARP-T1 WT but not ARP-T1 mutant suggesting that ARP-T1 is indeed phosphorylated by PKC and stable while ARP-T1 mutant, which lacks the phosphorylation site, is not phosphorylated by PKC (Fig. 2j).

**ARP-T1 interacts with proteins involved in ciliogenesis.** To understand the molecular pathways exploited by ARP-T1, we performed co-immunoprecipitation followed by mass spectrometry (MS) analysis to identify proteins interacting with ARP-T1. We transduced ACTRT1 (WT), ACTRT1 547_548insA (M) and vector control (V) tagged with DDK into the cells and immunoprecipitated using anti-FLAG M2 beads. We found putative interacting proteins in HeLa cells (Table 1, Fig. 3a), differentiated NHEK (Table 1), and differentiated hTERT-RPE1 cells (Table 1, Fig. 3b). ARP-T1 interacts in all cells with proteins involved in the primary cilium structure. We confirmed these interactions by immunoblot analysis with acetylated-tubulin, TCP8 (one of 8 subunits of chaperonin-containing T-complex), HSC70, BAG2, gamma-tubulin, EHD4, septin 2, and septin 9 antisera (Fig. 3a, b). While chaperones are often identified to some extent in MS analysis of precipitates with overexpressed proteins, the high number of spectral hits (Table 1) and the location of septins and EHD4 at ciliary basal body led us to observe the localization of ARP-T1 in ciliated hTERT-RPE1 cells.

We analyzed ciliogenesis, which was induced by serum starvation and differentiation, in ARPE-19 and hTERT-RPE1 cells. ARP-T1 localizes to the ciliary base when we co-stained ARP-T1 with acetylated-tubulin, which stains ciliary axoneme. ARP-T1 co-localized with rootletin, a major basal body protein, in primary cilium (Fig. 3c and Supplementary Fig. 1d in HaCaT cells). ARP-T1 interacts and forms a complex with gamma-tubulin, which localizes in the vicinity of the basal body (Fig. 3d and Supplementary Fig. 1e in HaCaT cells). We also confirmed that ARP-T1 co-localizes with EHD4 and septin 2 (Fig. 3d). We used proximity ligation assays to further confirm these interactions (Supplementary Fig. 1f, g). To reinforce these results, we chose to overexpress EHD4 and then locate ARP-T1. We transfected hTERT-RPE1 cells with EHD4 (EHD4) and vector control (V) tagged with V5. After immunoprecipitation using anti-V5 beads and immunoblot, we detected ARP-T1 in the EHD4 samples but not in the vector control (Supplementary

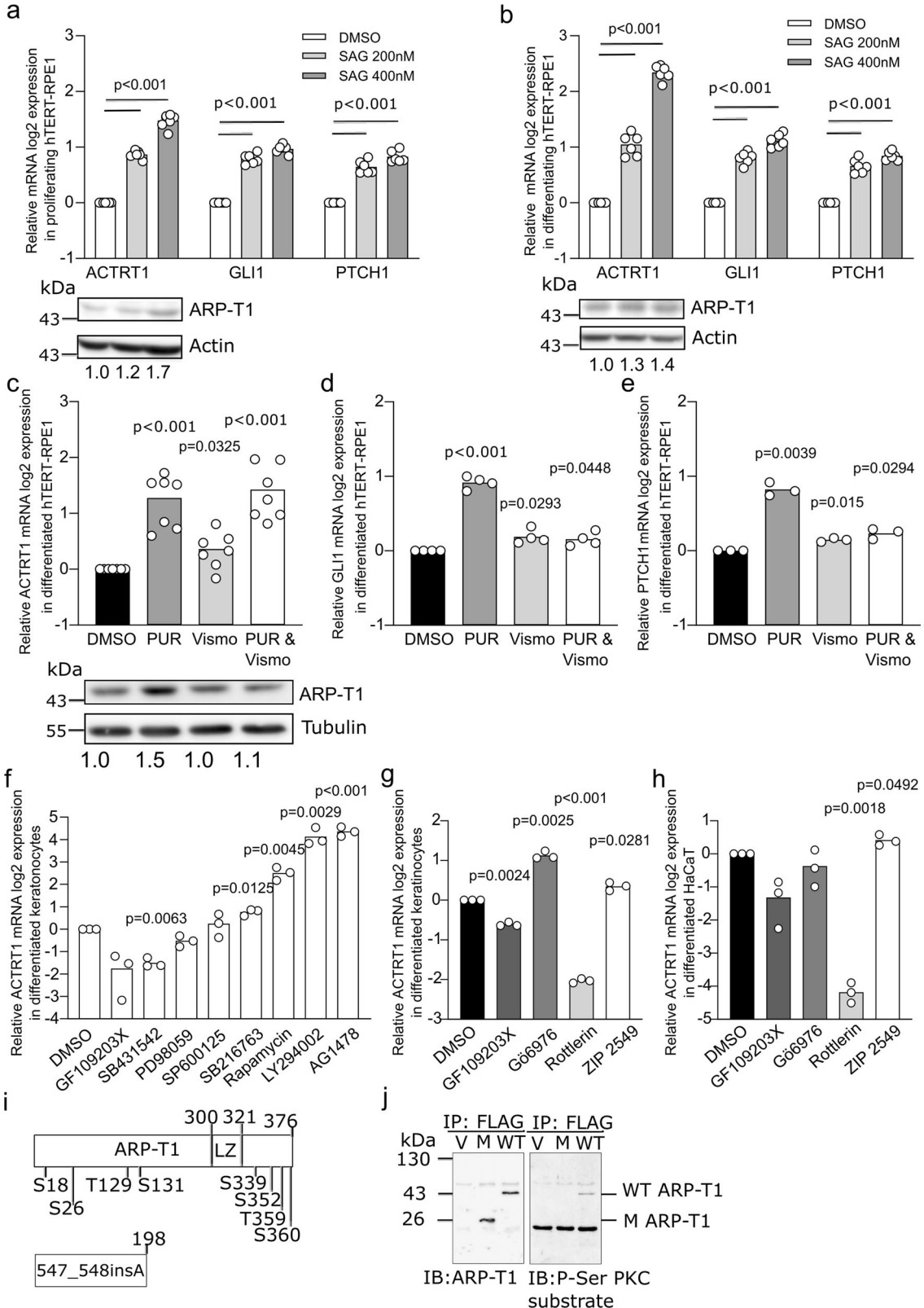

Fig. 1h). In the same cells, we confirmed that ARP-T1 co-localizes with EHD4 (Supplementary Fig. 1i).

**ARP-T1-associated BDCS tissues have shorter cilia.** Based on the localization of ARP-T1 to the basal body and its interaction with septin 2, which is involved in ciliary length control[24], we

analyzed the ciliary structures in the samples from BDCS patients (*ACTRT1 547_548insA*, mutations B2, mutation A3, mutation CNE12)[6] comparing to normal hair follicles and samples from a sporadic BCC patient. First, we observed the ciliary structures by staining with rootletin and acetylated-tubulin (Fig. 4a, top) and confirmed the co-localization of ARP-T1 and rootletin (Fig. 4a bottom, individual stainings with rootletin and acetylated-

**Fig. 2 ACTRT1 is regulated by non-canonical Hedgehog signaling pathway and by protein kinase C delta. a, b** Relative ACTRT1, GLI1, and PTCH1 mRNA expression (**a, b** $N = 6$) and ARP-T1 expression (**a** $N = 4$; **b** $N = 3$) upon treatment with SAG in hTERT-RPE1 cells under proliferative (**a**) and differentiating (**b**) conditions. Actin is used as loading control for ARP-T1 expression. Numbers under the blots present the fold change expression of ARP-T1 compared to the vehicle (DMSO) treatment. **c–e** Relative ACTRT1 (**c** $N = 7$), GLI1 (**d** $N = 4$), PTCH1 (**e** $N = 3$) mRNA expression and ARP-T1 expression (**c** $N = 4$) upon treatment with purmorphamine (PUR) and/or vismodegib (VISMO) in differentiated hTERT-RPE1 cells. Tubulin is used as loading control for ARP-T1 expression. Numbers beneath the blots represent the fold change expression of ARP-T1 compared to the vehicle treatment. **f, g** Relative ACTRT1 mRNA expression upon treatment with different protein kinase inhibitors (**f**) or with PKC inhibitors (**g**) in differentiated NHEK ($N = 3$). **h** Relative ACTRT1 mRNA expression upon treatment with PKC inhibitors in differentiated HaCaT cells ($N = 3$). **a–h** Data are presented as means of the fold change compared to the value of vehicle-treated samples. Each open circle represents one independent experiment. **i** Schematic representation of ARP-T1 and ARP-T1 mutant (547_548insA) with predicted phosphorylation sites. **j** ARP-T1 and Phospho-Serine (P-Ser) PKC expression in NHEK transduced with lentiviral vectors, empty vector (V), *ACTRT1* mutant (M), and *ACTRT1* WT (WT), after immunoprecipitated (IP) with anti-FLAG monoclonal antibody M2-conjugated agarose.

**Table 1 Proteins interacting with ARP-T1.**

| HeLa cells | ARP-T1 | 547_548 InsA | Vector |
|---|---|---|---|
| ARP-T1 | 183 | 91 | 0 |
| Heat shock cognate 71 kDa protein | 121 | 68 | 9 |
| T-complex protein 1 subunit theta (TCP1 theta) | 66 | 20 | 0 |
| RuvB-like 2 | 73 | 39 | 1 |
| Erlin-2 | 28 | 16 | 0 |
| BAG family molecular chaperone regulator 2 (BAG 2) | 91 | 33 | 4 |
| Ras-related protein Rab-8A | 7 | 4 | 0 |
| Dynactin | 19 | 3 | 0 |
| Importin subunit α-1 | 18 | 7 | 0 |
| Exportin 1 | 19 | 2 | 0 |
| Filamin-A | 11 | 5 | 0 |
| **NHEK cells** | | | |
| ARP-T1 | 139 | 104 | 0 |
| Heat shock 70 kDa protein | 38 | 19 | 0 |
| BAG family molecular chaperone regulator 6 (BAG 6) | 13 | 6 | 0 |
| T-complex protein 1 subunit beta (TCP1 beta) | 9 | 0 | 0 |
| Erlin-2 | 8 | 3 | 0 |
| Tubulin-6 | 7 | 3 | 0 |
| Rab18 | 7 | 2 | 0 |
| BAG 2 | 5 | 1 | 0 |
| **hTERT-RPE1 cells** | | | |
| BAG2 | 45 | 14 | 1 |
| TCP1 theta | 19 | 0 | 0 |
| RuvB-like 1 | 14 | 4 | 0 |
| EH domain containing protein 4 (EHD4) | 16 | 1 | 0 |
| Septin-2 | 12 | 2 | 1 |
| Septin-9 | 6 | 1 | 0 |
| Rab10 | 10 | 3 | 1 |
| PDZ and LIM domain protein 5 (PDLI5) | 10 | 0 | 0 |
| Rab6A | 5 | 0 | 0 |
| EH domain-containing protein 1 (EHD1) | 5 | 0 | 0 |
| Actin-related protein 2 | 4 | 1 | 0 |
| Cortactin | 11 | 9 | 4 |

tubulin, and rootletin and ARP-T1 are shown in Supplementary Figs. 2 and 3, respectively). Strikingly, the ciliary length (Fig. 4b) and fluorescence intensity of rootletin and ARP-T1 (Fig. 4c, d) were significantly reduced in the BDCS samples compared to normal hair follicles and sporadic BCC. Ciliary length, rootletin fluorescence and ARP-T1 fluorescence were particularly reduced

in *ACTRT1 547_548insA* (insA in Fig. 4b–f) and B2 mutation while less prominently reduced in mutations A2 and CNE12. We also found significant correlations between ARP-T1 fluorescence and ciliary length (Fig. 4e) and between ARP-T1 fluorescence and rootletin fluorescence (Fig. 4f). These results suggest that ARP-T1 is important for cilia length and that ARP-T1-associated BDCS might originate from ciliary defects.

To study the role of ARP-T1 in controlling ciliary length in vitro, we overexpressed *ACTRT1* WT and *ACTRT1 547_548insA* in hTERT-RPE1 cells. The length of primary cilia was quantified based on acetylated-tubulin staining. ARP-T1 mutant overexpression reduced the ciliary length, approximately by 25% (Fig. 4g and quantified in Fig. 4h, individual stainings are in Supplementary Fig. 4a). We then investigated more closely the role of ARP-T1 with ACTRT1 silencing in hTERT-RPE1 cells. Two different ACTRT1 shRNAs efficiently knocked down the expression of ACTRT1 at mRNA level (65% for KD1 and 50% for KD2; shown in Supplementary Fig. 4b) and at ARP-T1 level (30%, Fig. 4k and Supplementary Fig. 4c). Primary cilia in ACTRT1-depleted cells were shorter than in control cells (25% reduction; Fig. 4i, j and Supplementary Fig. 4d). These results were confirmed in HaCaT cells, in which ciliary length was halved with ACTRT1 KD1 and KD3 (Supplementary Fig. 5). To further confirm the role of ARP-T1 mutant in the control of ciliary length, we expressed *ACTRT1 547_548insA* resistant shRNA (MshR) in control and ACTRT1 KD cells. ARP-T1 MshR expression leads to an identical reduction of ciliary length in control and KD cells (shown by the quantification in Fig. 4l, stainings are shown in Supplementary Fig. 6a). These results confirmed that ARP-T1 loss of function and ACTRT1 silencing both lead to the reduction of the ciliary length. Therefore, ARP-T1 is required to maintain the normal length of cilia in vitro. In summary, ARP-T1 is located in ciliary basal body and regulates ciliogenesis both in vivo and in vitro. The absence of ARP-T1 gives rise to an epidermal ciliopathy.

**ARP-T1 loss of function disrupts septin 2 localization to cilia without affecting the actin cellular network.** Primary cilium is essential for HH pathway, and we described that ARP-T1 is localized at its basis and is necessary for the full length of the axoneme. Thus, we investigated the localization of ARP-T1 and its interacting proteins during HH activation. We found that ARP-T1 remains at the cilia basal body upon SMO activation, with SAG and purmorphamine, according to the co-staining with rootletin and septin 2 (Fig. 4m, individual stainings with rootletin and ARP-T1, and septin 2 and ARP-T1 are shown in Supplementary Fig. 6b, c, respectively). We also showed that remaining ARP-T1, in the ACTRT1 KD cells, still co-localizes with the rootletin, and septins 2 and 9 (Supplementary Fig. 6d) and that septin 9 localized to the axoneme, septin filaments and the base of the primary cilium in control and ACTRT1 KD cells under

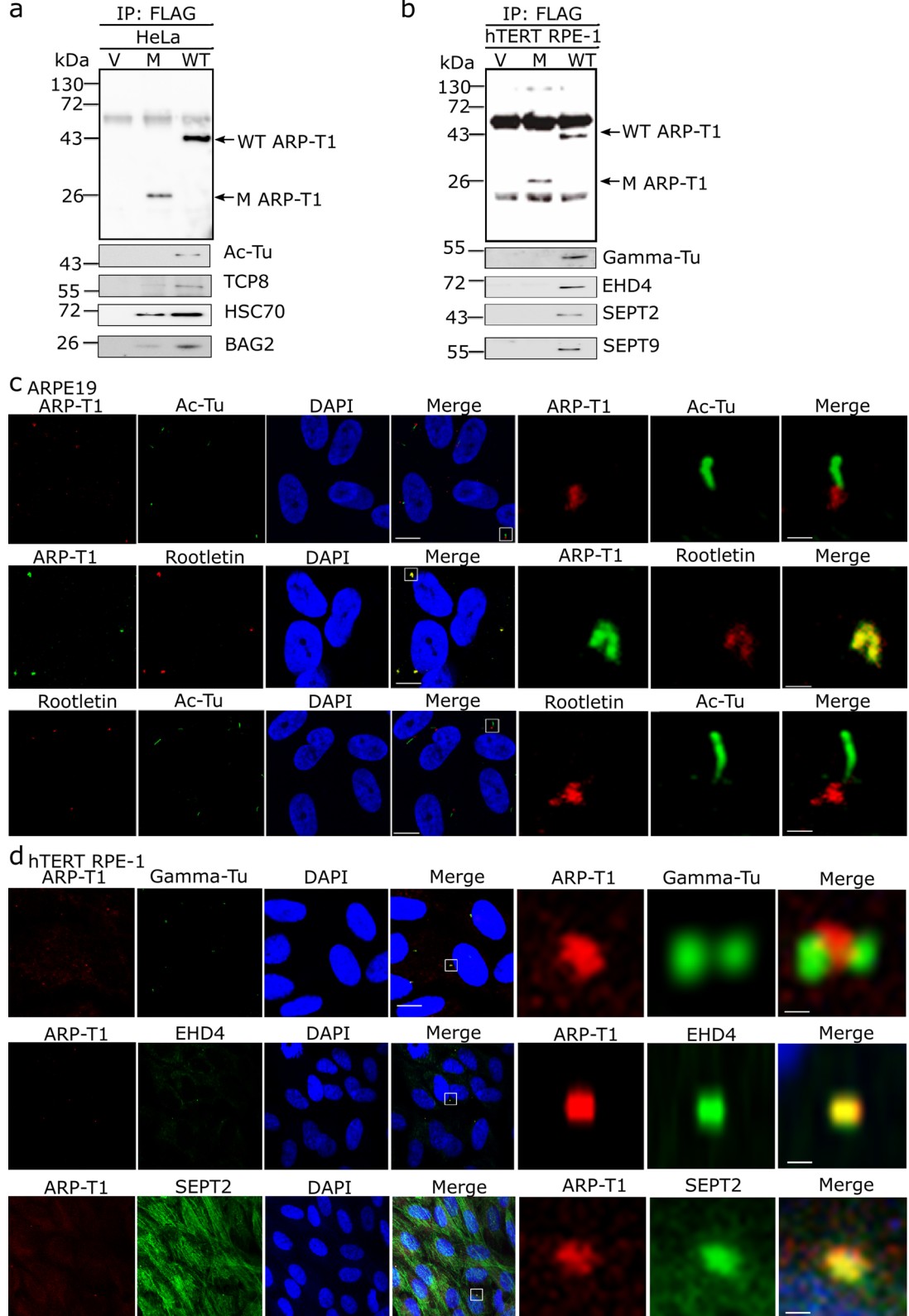

**Fig. 3 ARP-T1 interacts with proteins involved in ciliary machinery. a, b** HeLa (**a**) and hTERT-RPE1 (**b**) cells were transduced with lentiviral vectors, empty vector (V), *ACTRT1* mutant (M) and *ACTRT1* WT (WT), and immunoprecipitated (IP) with anti-FLAG monoclonal antibody M2-conjugated agarose, and analyzed by immunoblot with indicated antisera. **c** Immunofluorescence stainings of ARP-T1, acetylated-tubulin and rootletin in 35 days of serum-starved ARPE19 cells. Nuclei are stained with DAPI. Scale bar, 5 μm. Higher magnifications of the boxed area are shown on right three panels. Scale bar, 1 μm. **d** Immunofluorescence staining of ARP-T1, gamma-tubulin, EHD4, and septin 2 in 48 h of serum-starved hTERT-RPE1 cells. Nuclei are stained with DAPI. Scale bar, 5 μm. Higher magnifications of the boxed area are shown on the right three panels. Scale bar, 1 μm.

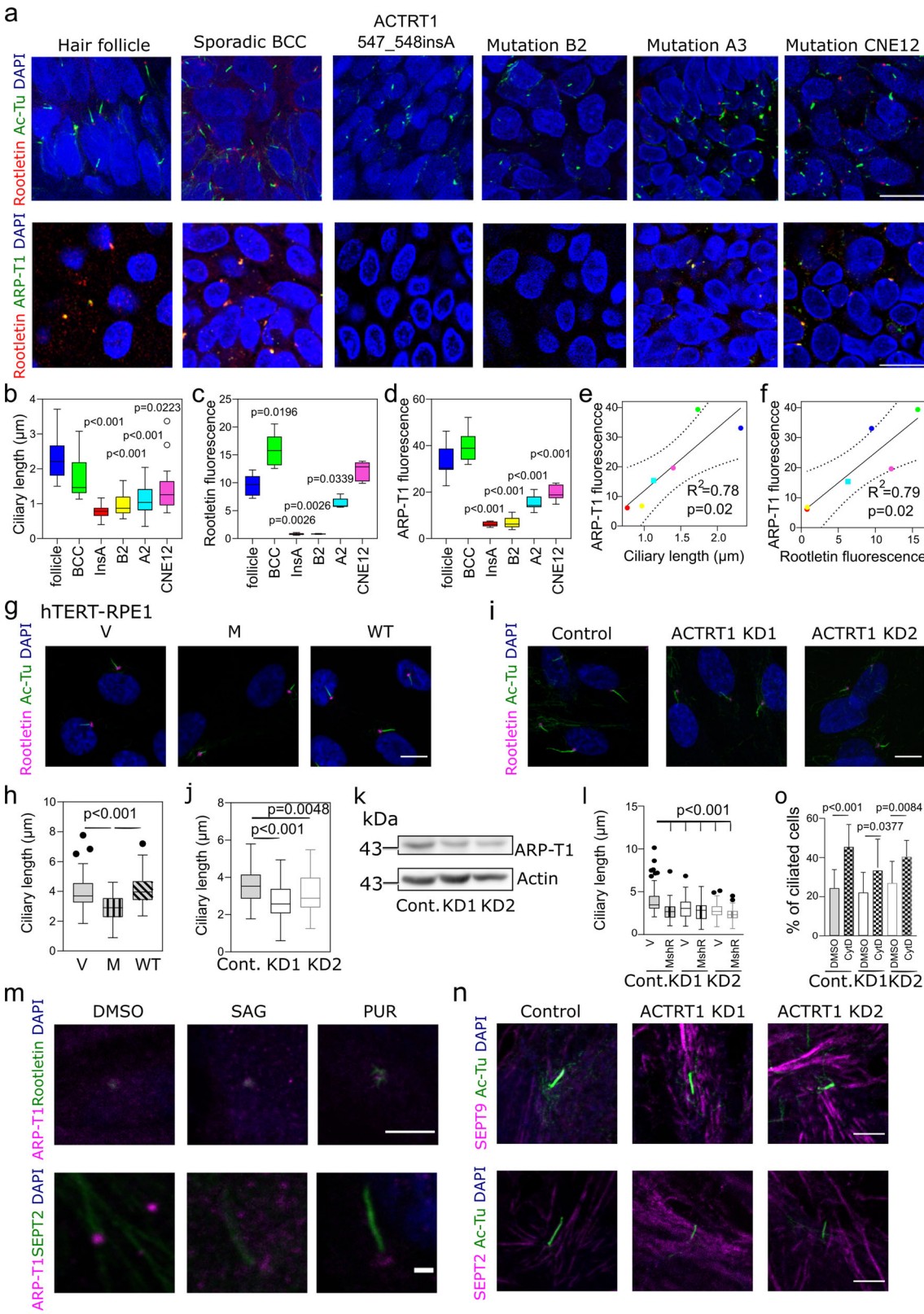

differentiation (Fig. 4n top, individual stainings in Supplementary Fig. 6e, quantification in Supplementary Fig. 6g). Therefore, ARP-T1 decrease does not affect neither its interactions with rootletin and septins nor septin 9 localization. Unexpectedly, we observed that septin 2, normally localized in the axoneme under serum starvation, is no longer there in ACTRT1 KD cells (Fig. 4n bottom, individual stainings in Supplementary Fig. 6f, quantification

in Supplementary Fig. 6h). We were able to detect septin 2 in its other locations, i.e. septin filaments and the base of the primary cilium.

Finally, as the actin cytoskeleton network is involved in ciliogenesis and as we found interactions between ARP-T1 and several proteins associated with this network (cortactin, septins, ARP2), we studied the actin cytoskeleton in ACTRT1 KD cells and its involvement in

**Fig. 4 The Bazex–Dupré–Christol syndrome is a ciliopathy caused by ARP-T1 loss of function, and knockdown of ACTRT1 in hTERT-RPE1 cells induces resorption of primary cilia. a** Representative immunofluorescence images using acetylated-tubulin (green) and rootletin (red) (top), and ARP-T1 (green) and rootletin (red) (bottom) in hair follicle, sporadic BCC and 4 BDCS (*ACTRT1 547_548insA*, mutation B2, mutation A3, mutation CNE12[6]). Cell nuclei are stained with DAPI (blue). Scale bar, 5 µm. **b** Quantification of the ciliary length from 3D confocal immunofluorescence microscopy images. **c, d** Quantification of the relative fluorescence intensity of rootletin (**c** $N = 5$) and ARP-T1 (**d** $N = 10$) on the ciliary rootlet. **e, f** Correlation between the ARP-T1 fluorescence and ciliary length (**e**), and between the ARP-T1 and rootletin fluorescence (**f**). **g** Immunofluorescence stainings of acetylated-tubulin (green) and rootletin (pink) in 48 h serum starved hTERT-RPE1 cells expressing an empty vector (V), or *ACTRT1* mutant (M), or *ACTRT1* WT (WT). Cell nuclei are stained with DAPI (blue). Scale bar, 10 µm. **h** Quantification of ciliary length of (**g**). **i** Immunofluorescence stainings of acetylated-tubulin (green) and rootletin (pink) in 48 h serum-starved control and ACTRT1 KD hTERT-RPE1 cells. Cell nuclei are stained with DAPI (blue). Scale bar, 10 µm. **j, k** Quantification of the ciliary length (**j**) and ARP-T1 protein level (**k** $N = 3$) in control (Cont.) and ACTRT1 KD hTERT-RPE1 cells. **l** Quantification of ciliary length in 48 h serum-starved control and ACTRT1 KD hTERT-RPE1 cells expressing an empty vector (V), or *ACTRT1* mutant resistant to shRNA (MshR). **b–d, h, l** Results are presented as Tukey box-plot. Black circles represent outliers. **m** Representative immunofluorescence stainings of ARP-T1 (pink) and rootletin (green, top) or septin 2 (green, bottom) upon treatment with SAG or purmorphamine (PUR) in hTERT-RPE1 cells under differentiating condition. Scale bar, 5 µm (top) or 1 µm (bottom). **n** Immunofluorescence stainings of acetylated-tubulin (green) and septin 9 (pink, top) or septin 2 (pink, bottom) in 48 h serum-starved control and ACTRT1 KD hTERT-RPE1 cells. Scale bar, 1 µm. **o** Percentage of ciliated control and ACTRT1 KD hTERT-RPE1 cells under proliferative condition, after treatment with cytochalasin D (CytD). Data are presented as means of the percentage ± SD.

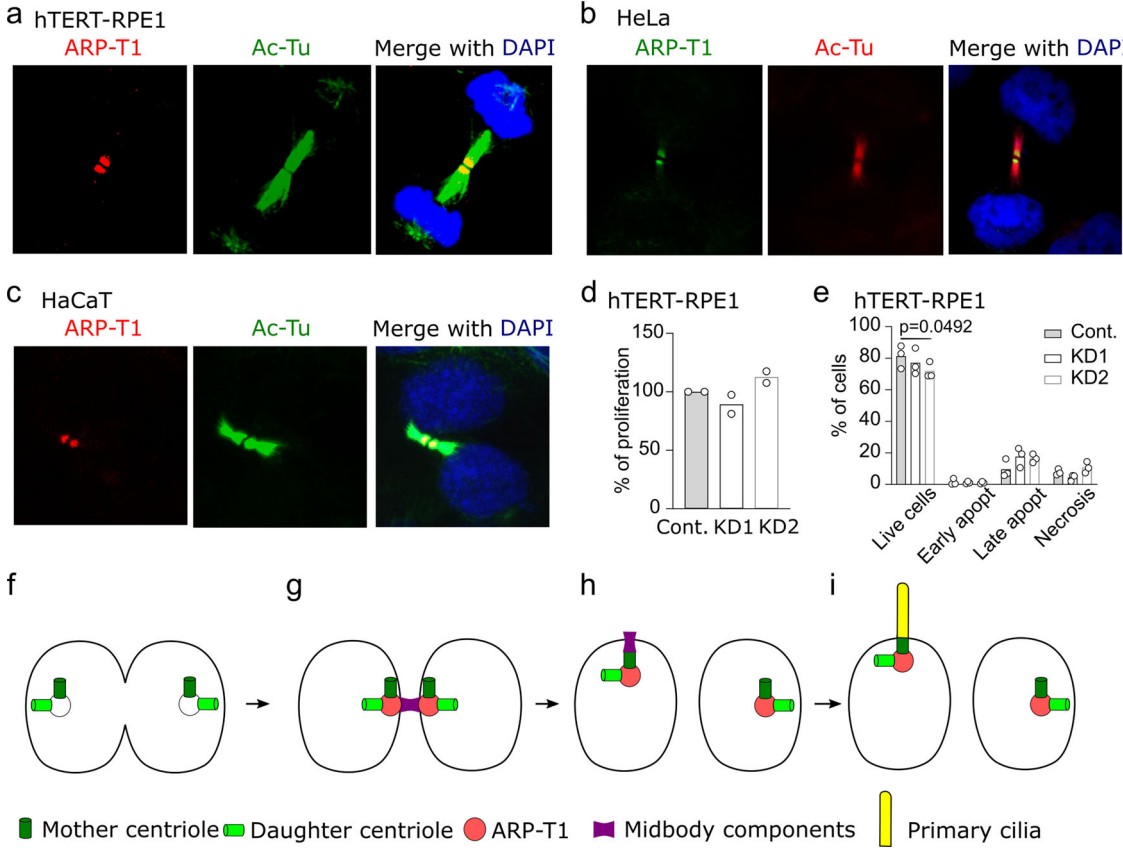

**Fig. 5 ARP-T1 localizes to midbody during cytokinesis. a** Immunofluorescence stainings of ARP-T1 (red) and acetylated-tubulin (green) in hTERT-RPE1 cells. Cell nuclei are stained with DAPI (blue). **b** Immunofluorescence stainings of ARP-T1 (green) and acetylated-tubulin (red) in HeLa cells. Cell nuclei are stained with DAPI (blue). **c** Immunofluorescence stainings of ARP-T1 (red) and acetylated-tubulin (green) in HaCaT cells. Cell nuclei are stained with DAPI (blue). **d, e** Proliferation (**d**) and apoptosis (**e**) analyses of control (Cont.) and ACTRT1 KD hTERT-RPE1 cells. Data are presented as means of the percentage. Each open circle represents one independent experiment. **f–i** Model for ARP-T1 traveling from midbody to the primary cilium.

cilia formation. We did not find any difference in actin filaments comparing ACTRT1 KD and control cells (Supplementary Fig. 6i). Moreover, cytochalasin D-induced ciliogenesis[25] was not blocked by silencing of ACTRT1 (Fig. 4o), indicating that the actin cytoskeletal network is not deficient in these cells and is unlikely to be the cause of ciliary shortening.

**ARP-T1 also localizes to the midbody in dividing cells.** It was reported that septin 2 acts as a scaffold for myosin II and its kinases at the cleavage furrow[26] and that septin 9 mediates

midbody abscission during cytokinesis[27]. Therefore, we searched for the localization of ARP-T1 in dividing cells. ARP-T1 localized at the midbody in hTERT-RPE1 cells (Fig. 5a), HeLa cells (Fig. 5b) and HaCaT cells (Fig. 5c). To determine whether ARP-T1 was implicated in cell division, we performed proliferation and apoptosis assays. Using EdU, an analog of thymidine, we found that ARP-T1 is not necessary for proliferation because control and ACTRT1 KD cells incorporated identical amounts of EdU (Fig. 5d). We then analyzed apoptosis of these cells. ARP-T1 also has no effect on the different stages of apoptosis, although we

observed a slight decrease in the live cell population with KD2 (Fig. 5e). In our conditions, ACTRT1 silencing has no obvious impact on cell division. Postmitotic midbody is directly implicated in primary ciliogenesis. After cytokinesis, the midbody remnant moves along the apical surface, proximal to the centrosome at the center of the apical surface where the primary cilium emerges. The midbody remnant carries RAB8, IFT20, IFT88 and exocyst subunits required for ciliogenesis. If the remnant is removed, primary ciliogenesis is greatly impaired[28]. Midbody remnant licenses primary cilium formation[29]. Both midbodies and primary cilia contain acetylated-tubulin and several proteins, which have been found to relocate to the base of assembling cilia and to participate in ciliogenesis and in cilia-mediated signaling[30] (Fig. 5f–i).

## Discussion

Here, we report that ARP-T1 localizes to the basal body of primary cilium where it is involved in the regulation of the extension of the ciliary axoneme. BDCS tumors and ACTRT1-deficient cells show shortened primary cilia. Thus, BDCS is a novel and first ciliopathy implicated in skin cancer caused by mutations of ACTRT1 or its enhancer RNA elements. ARP-T1 supports intact cilia and controls proper ciliogenesis, potentially through septin 2 involvement. ARP-T1 is highly expressed during epidermal differentiation, is stabilized by protein kinase C and ACTRT1 expression is regulated by PKC delta and by a non-canonical HH pathway.

ACTRT1/ARP-T1 regulation has not been described yet. Here, we show that HH and mTORC2/PKC delta pathways positively regulate ACTRT1 on mRNA level. Upon cell differentiation, HH signaling is activated at the cilium and known to induce canonical and non-canonical signaling pathways. As ARP-T1 was previously reported to activate HH signaling[6], it was important to decipher a possible feedback loop with this pathway. We found that SMO activation favors ACTRT1 mRNA and ARP-T1 protein expression. However, a blockade with vismodegib, used in the clinic to treat BCC patients, could not reverse purmorphamine-induced ACTRT1 mRNA expression. We concluded that ACTRT1 mRNA is regulated by non-canonical HH signaling. We next unraveled which phosphorylation pathway was implicated in this non-canonical HH signaling. MAP kinase inhibition with PD98059 had no effect on ACTRT1 mRNA expression. By using PI3K/mTORC1/S6K and mTORC2/PKC delta inhibitors, we found that the proliferating signaling PI3K/mTORC1/S6K limits ACTRT1 mRNA expression while its differentiating counterpart mTORC2/PKC delta[31] favors it. These findings highlight the role of HH and mTORC2/PKC delta pathways in ACTRT1/ARP-T1 regulation.

By MS analysis, we identified a machinery of molecular ARP-T1 interactors involved in primary cilium formation (see Table 1): (1) Chaperon containing TCP-1 (CCT/TRiC), consists of two identical stacked rings, each containing eight subunits (TCP1-8). This complex folds various proteins including actin and tubulin. It was shown that CCT/TRiC family chaperonin localizes to centrosomes, forms a complex with BBS proteins BBS6, BBS10, and BBS12, and mediates the assembly of the BBSome, involved in ciliogenesis regulating transports of vesicles to the cilia[32]. This BBSome protein complex binds to Rabin 8, the GTP/GDP exchange factor, for the small GTPase RAB8. RAB8 localizes to the cilium, is required for ciliogenesis, and mediates the docking and fusion of vesicles. It was proposed that the BBSome acts upstream of RAB8 in this vesicular transport to the cilium[33]. BBS causing IFT27 mutations point to a link of the BBSome with IFT where this small GTPase might be involved in delivering cargo from the BBSome to the IFT[34] (2) CCT and

HSC70 form a stable complex and this interaction was suggested to be used to deliver the unfolded substrate from HSC70 to the substrate-binding region of CCT[35]. HSC70 interacts with IFT88[36]. Bcl2-associated athanogen (BAG2) protein forms a chaperone complex with HSC70 and regulates their protein folding activity[37]; (3) The Eps15 homology domain (EHD) family of proteins, composed of EHD1-4, is associated with RAB8 and RAB11 membranes and regulates endosomal membrane trafficking[38]. EHD4 locates in the transition zone of cilia and plays a role in protein transport to cilia and ciliogenesis. Ehd4 deficient mice show a strong phenotype in skin, kidney, and testis with small testis, reduced sperm motility, sperm number, and germ cells, such that Ehd4-/- mice are subfertile[39]; (4) Septin 2 localizes to the base of the axoneme. Cells completely depleted of septin 2 lack a cilium whereas cells partially depleted at the base of primary cilium show a significantly shortened cilium as compared to controls[24]. Septin 2 interacts with HSC70[40]. Septin 9 has been shown to interact with microtubules[41] and localizes to the ciliary pocket region[42]. Primary cilia in septin 9 depleted cells are shorter than those present in control cells[42]. Gamma-tubulin associates with centrosomes and localizes to the vicinity of basal body[43].

Overall, our MS analysis results point to a function of ARP-T1 in the ciliary basal body at the crossroads of vesicle trafficking, protein folding, and the cytoskeleton (Supplementary Table 1). The ciliary pocket or periciliary PM serves as an interface with the actin cytoskeleton, which is important for the movement of the basal body to the PM and for vesicular and endosomal trafficking[10,44–46]. We hypothesize that ARP-T1 is functional at this interface and may stabilize the cytoskeletal matrix.

Functional genomic screening showed that proteins involved in actin dynamics and endocytosis are important for ciliogenesis. Depletion of gelsolin family proteins (GSN and AVIL), which regulate cytoskeletal actin organization by severing actin filaments, reduced the number of ciliated cells and silencing of ACTR3 (ARP-3), a major constituent of the ARP2/3 complex involved in nucleating actin polymerization, increased cilium length[47]. On the other hand, it was reported that the actin cytoskeleton is disrupted in KD cell lines targeting ciliary proteins. For example, KD cells of Ahi1, whose human ortholog is mutated in Joubert syndrome, showed disorganized and decreased actin filaments[48]. ARP-T1 displays 49 % identity with β-actin and 40%, 37%, 29% identity with ARP-1, ARP-2, ARP-3, respectively[7]. ARP-T1 as a basal body protein is involved in a positive regulation of ciliogenesis. Despite all these clues, we could not detect a difference in the actin filaments between ACTRT1 KD and control cells, and our cytochalasin D-induced ciliogenesis experiments did not allow us to confirm a role of the actin cytoskeletal network in the shortened cilia observed in silenced ACTRT1 cells. Similar to ARP-T1, some cancer-associated mutations result in shorter cilia or faster ciliary resorption (1) Von Hippel-Lindau (VHL) disease is characterized by the development of premalignant renal cysts, and arises because of functional inactivation of VHL tumor suppressor protein (pVHL). pVHL maintains the structural integrity of the primary cilium for suppression of uncontrolled proliferation of kidney epithelial cells and cyst formation[49]. (2) It was shown by electron microscopy that glioblastoma cells and tumors contain immature primary cilia and basal body/centrioles[50,51]. (3) Mutations in Budding uninhibited by benzimidazole-related 1 (BUBR1), a molecule of spindle assembly checkpoint, cause premature chromatid separation (mosaic variegated aneuploidy), cancer disposition, and impaired ciliogenesis[52].

The transition zone or diffusion barrier is another cellular structure essential for the integrity of the primary cilium[41]. Septins are key components of the diffusion barrier at the base of

the primary cilium, forming ring-like structures and being transported to the axoneme. The involvement of septin 2 in ciliogenesis was particularly studied these last years[24,42,53]. In an attempt to understand how the knockdown of ACTRT1 led to a shortening of ciliary length, we observed that septin 2 was absent from the axoneme in ACTRT1 silenced cells, but preserved in septin filaments and in the base of the primary cilium. In contrast, the localization of septin 9 at the axoneme and the base of cilia and in septin filaments remained unchanged. These results suggest that the ciliogenesis defect observed in ACTRT1-deficient cells could be related to the mislocalization of septin 2, as it is required for normal axoneme extension[24]. Future experimental work will be needed to further investigate the link between septin 2 and ARP-T1 in cilia formation.

Septins are not only essential for the diffusion barrier at the base of the primary cilium, but also participate in cell division, where they are located in the midbody[27]. Our investigations have shown that ARP-T1 is also found in the midbody of dividing cells, and that it could therefore have an impact on cell division. Nevertheless, cell proliferation and apoptosis were not affected by the decrease of ARP-T1, and thus ARP-T1 at the midbody level has no effect on cell division. This was not unexpected, as Pejskova et al.[54] recently described that KIF14, a protein involved in ciliogenesis, was also localized in the midbody without any impact on cell division.

With our results, we propose a model where ARP-T1 contributes to primary ciliogenesis via direct interaction with proteins of the ciliary machinery. First, ARP-T1 is localized to the mitotic spindle through the interaction with gamma tubulin, which is present between the mother and the daughter centrioles (Fig. 5f). As cytokinesis proceeds, the complex of ARP-T1 and two centrioles moves to the central spindle where the midbody is formed by recruiting septins (Fig. 5g). The midbody cleaves one side of the intracellular bridge and remains tethered to one of the daughter cells. In the complex of ARP-T1, two centrioles and midbody move towards the apical surface of the daughter cells where the primary cilium will be formed (Fig. 5h, i).

Dynamic HH signaling has been identified in mammalian post-natal testis, and HH pathway components PTCH1, SUFU, and GLI1 were identified in spermatogonia, spermatocytes and spermatids[55], where spermatids express high levels of SUFU to repress HH activity[56]. It was previously reported that ARP-T1 binds to the GLI1 promoter and inhibits GLI1 mRNA expression and signaling[6]. In the current study, we showed that the elevated level of ACTRT1 mRNA with purmorphamine was not decreased by the treatment with purmorphamine and vismodegib (Fig. 2c–e). Nevertheless, this regulation is absent on ARP-T1 protein level, as purmorphamine-induced ARP-T1 expression was counteracted by the addition of vismodegib (Fig. 2c). Thus, we hypothesize that ACTRT1 functions in a SMO-independent HH pathway and that post-transcriptional modifications, as shown for PKC, regulate ARP-T1 function. The activation of a non-canonical HH pathway may explain the apparent paradox that we observed shortened primary cilia both in BCC from BDCS and in ACTRT1 KD cells. While the loss of primary cilia is observed in diverse cancers where it correlates with worse prognosis, their retention is paramount for medulloblastoma and BCC development[9]. Nevertheless, SMO-independent operation occurs both in medulloblastoma[12] and BCC mouse models[13], and was recently linked to the development of resistance to SMO-inhibition[57]. This suggests that BDCS is a model for the study of vismodegib resistance in BCC and medulloblastoma.

Functions of the primary cilia, including activation of the HH signaling, are dependent on the ciliary length and on proteins involved in ciliogenesis, such as proteins located in the basal body[58]. In addition to this, ARP-T1 was reported to regulate the HH pathway[6]. As we demonstrated that ACTRT1 silencing is associated with a shortened primary cilium, we assessed the expression of HH proteins in our deficient cells. GLI1 and PTCH1 mRNA expression tend to increase but not significantly in ACTRT1 KD cells (Supplementary Fig. 6j, k), suggesting that our silencing might be insufficient to correctly activate the HH signaling, or that other compensating elements might prevent a stronger activation of the pathway. In order to look for a second hit required for a strong activation of GLI1, we studied in depth the transcriptomic analysis performed in Bal et al.[6], and unraveled an approximately 17-fold increased expression of ARHGAP36 mRNA in BDCS samples. ARHGAP36 gene is located on the X chromosome, in close proximity to the ACTRT1 gene, and is known to activate the HH pathway in medulloblastoma[59]. ARHGAP36 might contribute to GLI1 activation in BDCS patients. Another gene of interest is SMARCA1. Indeed, SMARCA1 expression is often lost in gastric cancer cells due to methylation[60] and in soft tissue sarcomas[61]. This gene is located next to the ACTRT1 gene, and cannot be excluded that mutations in the RNA elements regulating ACTRT1 do not also affect the expression of SMARCA1, which expression is 1.8-fold higher in BDCS patients[6]. SMARCA1 encodes for the probable global transcription activator SNF2L1, which acts as a chromatin remodeler and interacts with actin. Loss of expression of SMARCA1 increased proliferation of gastric cancer cells, implying a tumor suppressor role[60]. A whole-exome sequencing of BDCS tumors is presently ongoing and we are currently studying the link between ARHGAP36, SNF2L1, and ARP-T1. These results will be the focus of a future publication.

Our results suggest that ARP-T1 is directly or indirectly involved in a non-canonical HH pathway connecting the actin cytoskeleton organization involved in vesicle transport, basal body formation, and the formation of the primary cilium to prevent the pathogenesis of BDCS. This study also sheds light on how ARP-T1-associated ciliary defects might contribute to carcinogenesis. ARP-T1 could be a new target for novel therapeutic approaches in BDCS and BCC, the most frequent human cancer. This appears particularly important since loss of primary cilia is a recognized mechanism of resistance to SMO inhibitors in medulloblastoma and BCC[57,62]. Finally, Glaessl et al.[63] described BDCS BCCs to be more aggressive and prone to relapse. As BDCS patients have a shortened primary cilium, we hypothesize that BDCS BCCs represent a more advanced form of BCC.

## Methods

**Cell culture and samples**. Normal human epidermal keratinocytes (NHEK, established from neonatal foreskin in our laboratory) and immortalized keratinocytes, HaCaT cells (Invitrogen, Basel, Switzerland), were grown in EpiLife medium (Gibco, Invitrogen) with Human Keratinocyte Growth Supplement (Gibco, Invitrogen), 10 µg/ml gentamicin and 0.25 µg/ml amphotericin B (Gentamicin/ Amphotericin B solution, Gibco, Invitrogen). Spontaneously immortalized adult retinal epithelial cell line 19 (ARPE-19) and immortalized hTERT-RPE1 cells (ATCC, Manassas, VA) were cultured in Dulbecco's Modified Eagle's Medium (DMEM) High Glucose, GlutaMAX™, Pyruvate (Gibco, Invitrogen) or DMEM/F12 (ATCC) with 0.01 mg/ml hygromycin B (Sigma-Aldrich, St Louis, MO), both media supplemented with 10% fetal bovine serum (FBS), 100 U/ml penicillin and 100 µg/ml streptomycin (BioConcept AG, Allschwil, Switzerland). HeLa and HEK293T (Invitrogen) were cultured in DMEM supplemented with 10% FBS, 100 U/ml penicillin and 100 µg/ml streptomycin. The cells were incubated at 37 °C in a 5% $CO_2$ atmosphere.

To induce cilia formation, NHEK and HaCaT were cultured up to 7 days in EpiLife media supplemented with 2 mM $CaCl_2$. hTERT-RPE1 and ARPE-19 cells were cultured 48 h or 35 days, respectively, in media containing 0.2% FBS.

BDCS samples were collected and prepared as previously described[6].

**Lentivirus production and transduction**. Lentivirus particles (LVs) for ACTRT1 shRNA were produced by Sigma-Aldrich, in pLKO backbone vector. ShRNA sequences are for ACTRT1 KD1: 5′ CCG GGC CTG GTT TCT ACC TGT CTA ACT CGA GTT AGA CAG GTA GAA ACC AGG CTT TTT G 3′, and ACTRT1 KD2: 5′ CCG GGT GCC TTT AGC AAG ACT TAA TCT CGA GAT

TAA GTC TTG CTA AAG GCA CTT TTT TG 3′; ACTRT1 KD3: CCG GCA TGA CCT CTA TGA GCA GTT TCT CGA GAA ACT GCT CAT AGA GGT CAT GTT TTT G. Control is an empty vector with same selection. LVs for ACTRT1 over-expression were produced with calcium phosphate transfection method: ACTRT1 wild-type (WT) and mutant containing the 547-548InsA mutation constructs were produced as previously described[6]. HEK293T cells were transiently co-transfected with psPAX2 (Addgene, Cambridge, MA) and pMD2.G (Addgene) and empty vector or ACTRT1 WT or mutant to produce LVs. LVs were harvested 48 h later and the titer was determined in HeLa cells[64]. ACTRT1 547-548InsA construct was mutated using QuikChange Multi Site-Directed Mutagenesis Kit (200514, Agilent Technologies, Basel, Switzerland) in the KD1 target (T599G, C602T, C605T) and reverse KD2 target (A341G, A344C, T556G) with following primers (Microsynth, Balgach, Switzerland): KD1 target GCCTGGGTTTTATCTGTCTAA, KD2 target GATTCAGTCTGGCCAAAGGCAC.

HeLa, hTERT-RPE1 and HaCaT cells were transduced using polybrene (8 ug/mL, TR1003, Millipore), with LVs O/N at 37 °C. The next day, LVs were removed and cells grew in their medium. When necessary, selection with puromycin (2.5 ug/mL, 540411, Calbiochem) or blasticidin (7.5 ug/mL, 15205, Sigma-Aldrich) started 24 h later.

**Drug treatments**. Cells were treated 24 h with SMO agonists 200 and 400 nM SAG (566660, Calbiochem) and 3 µM purmorphamine (4551, Tocris bioscience), or antagonist 5 µM vismodegib (S1082, Selleckchem), or kinase inhibitors: 10 µM GF109203X (ALX-270-049, Enzo Life Sciences), 10 µM SB431542 (1614, Tocris bioscience), 20 µM PD98059 (9900, Cell Signaling Technologies), 50 µM SP600125 (270-339-M005, Alexis Biochemicals), 10 µM SB216763 (1616, Tocris bioscience), 200 nM Rapamycin (R0161, LKT Laboratories), 50 µM LY294002 (70920, Cayman Chemical), 5 µM AG1478 (270-036-M001, Alexis Biochemicals), 3 µM Gö6976 (12060S, Cell Signaling Technologies), 10 µM Rottlerin (1610, Tocris bioscience), 20 µM ZIP (2549, Tocris bioscience, kind gift from Prof. Anthony Oro, Stanford University School of Medicine).

To induce cilia in proliferating condition, hTERT-RPE1 cells were treated for 16 h with 50 nM cytochalasin D (250255, MERCK).

**Western blot**. NHEK, HaCaT, ARPE-19, and hTERT-RPE1 cells were harvested in ice-cold FLAG (50 mM Tris/HCl pH 7.5, 150 mM NaCl, 1 mM EDTA, 1% Triton X-100, and protease inhibitor cocktail [Complete MiniTM tablette, Roche Diagnostics]) or RIPA (50 mM Tris/HCl pH 7.4, 150 mM NaCl, 12 mM Na deoxycholate, 1% NP-40, 0.1% SDS, protease inhibitor cocktail, and phosphatase inhibitor cocktail [PhosSTOP, PHOSS-RO, Roche]) lysis buffers for 30 min on ice with extensively pipetting every 10 min, then spun for 10–20 min at 12,000 × g at 4 °C. Supernatants were collected, heated for 5 min at 85 °C and centrifuged. Total protein concentration was determined by BCA assay (PierceTM BCA Protein Assay Kit; Thermo Scientific, Waltham, MA, USA). Fifteen ug of proteins were loaded onto a 10% SDS-PAGE gel and electroblotted onto a nitrocelulose or PVDF membrane (Hybond ECL; Amersham, UK). Membranes were incubated overnight at 4 °C with the following antibodies: anti-ARP-T1 (1:1000, SAB1408334, Sigma-Aldrich/1:2000, GP-SH6, Progen), anti-keratin 10 (1:200, MS-611-P0, Thermo Scientific), anti-IFT88 (1:1000, 13967-1-AP, Proteintech, UK), anti-actin (loading control, 1:5000, A2066, Sigma-Aldrich) and anti-alpha tubulin (loading control, 1:5000, T9026, Sigma-Aldrich). After washes in TBS-T, 1 h incubation at room temperature (RT) with HRP-secondary antibodies (anti-mouse [1:5000, NA931V, GE Healthcare UK Limited], anti-rabbit [1:5000, NA934V, GE Healthcare], anti-guinea pig [1:5000, ab97155, Abcam]), and revelation with chemiluminescence (ECL Prime, Amersham/WesternBright Quantum, Advansta, Witec, Sursee, Switzerland), images were acquired using the Luminescent Image Analyzer LAS-4000 mini (Fujifilm, Tokyo, Japan). Band intensity was analyzed with Image J software (National Institutes for Health, USA).

**Co-immunoprecipitation**. Transduced HeLa, NHEK, and serum-starved hTERT-RPE1 cells were washed with PBS and harvested in the FLAG lysis buffer. Supernatants were incubated with 30 µL anti-FLAG-beads (ANTI-FLAG™ M2 Affinity Gel) overnight at 4 °C and 1 h at RT on a rotator to pool-down ARP-T1-FLAG proteins. Beads were separated by centrifugation for 3 min at 5000 × g, washed five times with TBS, and eluted in 23 µL TBS and 7 µL 5x SDS-sample buffer at 85 °C for 5 min. ARP-T1 precipitation was confirmed using ARP-T1 antiserum (1:2000, GP-SH6), and co-precipitated proteins were analyzed using anti-acetylated tubulin (1:1000, T6793, Sigma-Aldrich), anti-TCP8 / TCP1 theta (1:500, PA5-30403, Thermo Scientific), anti-HSC70 (1:200, PA5-27337, Thermo Scientific), anti-BAG2 (1:100, PA5-30922, Thermo Scientific), anti-gamma tubulin (1:500, ab11316, Abcam), anti-EDH4 (1:1000, Dr. Plomann's lab), anti-septin 2 (1:2000, HPA018481, Sigma-Aldrich), anti-septin 9 (1:2000, HPA042564, Sigma-Aldrich).

**Proteomic analysis/mass spectrometry**. Transduced HeLa, NHEK, and serum-starved hTERT-RPE1 cells were washed with PBS and harvested in FLAG lysis buffer, and analyzed by the Proteomic Analysis Facility of University of Lausanne with the following protocol.

*Gel separation and protein digestion*. Protein samples were loaded on a 12% mini polyacrylamide gel and migrated about 2.5 cm in non-reducing conditions. After Coomassie staining, gel lanes between 15–300 kDa were excised into 5-6 pieces, and digested with sequencing-grade trypsin (Promega) as described by Shevchenko and colleagues[65]. Extracted tryptic peptides were dried and resuspended in 0.05% trifluoroacetic acid, 2% (v/v) acetonitrile for mass spectrometry analyses.

*Mass spectrometry analyses*. Tryptic peptide mixtures were injected on a Dionex RSLC 3000 nanoHPLC system (Dionex, Sunnyvale, CA, USA) interfaced via a nanospray source to a high-resolution mass spectrometer based on Orbitrap technology: Orbitrap Fusion Tribrid or QExactive Plus instrument (Thermo Fisher, Bremen, Germany), depending on the experiments considered. Peptides were loaded onto a trapping microcolumn Acclaim PepMap100 C18 (20 mm × 100 µm ID, 5 µm, Dionex) before separation on a C18 reversed-phase analytical nano-column, using a gradient from 4 to 76% acetonitrile in 0.1% formic acid for peptide separation (total time: 65 min).

Q-Exactive Plus instrument was interfaced with an Easy Spray C18 PepMap column (25 cm × 75 µm ID, 2 µm, 100 Å, Dionex). Full MS survey scans were performed at 70,000 resolution. In data-dependent acquisition controlled by Xcalibur software (Thermo Fisher), the 10 most intense multiply charged precursor ions detected in the full MS survey scan were selected for higher energy collision-induced dissociation (HCD, normalized collision energy NCE = 27%) and analysis in the orbitrap at 17'500 resolution. The window for precursor isolation was of 1.5 $m/z$ units around the precursor and selected fragments were excluded for 60 s from further analysis.

Orbitrap Fusion Tribrid instrument was interfaced with a reversed-phase C18 Nikkyo column (75 µm ID × 15 cm, 3.0 µm, 120 Å, Nikkyo Technos, Tokyo, Japan) or a custom packed column (75 µm ID × 40 cm, 1.8 µm particles, Reprosil Pur, Dr. Maisch). Full survey scans were performed at a 120,000 resolution, and a top speed precursor selection strategy was applied to maximize the acquisition of peptide tandem MS spectra with a maximum cycle time of 3 s. HCD fragmentation mode was used at a normalized collision energy of 32%, with a precursor isolation window of 1.6 $m/z$, and MS/MS spectra were acquired in the ion trap. Peptides selected for MS/MS were excluded from further fragmentation during 60 s.

*Data analysis*. MS data were analyzed using Mascot 2.5 (Matrix Science, London, UK) set up to search the Swiss-Prot (www.uniprot.org) database restricted to Homo sapiens taxonomy (UniProt, December 2015 version: 20'194 sequences). Trypsin (cleavage at K,R) was used as the enzyme definition, allowing two missed cleavages. Mascot was searched with a parent ion tolerance of 10 ppm and a fragment ion mass tolerance of 0.50 Da (Orbitrap Fusion) or 0.02 Da (QExactive Plus). Carbamidomethylation of cysteine was specified in Mascot as a fixed modification. N-terminal acetylation of protein and oxidation of methionine were specified as variable modifications.

Scaffold software (version 4.4, Proteome Software Inc., Portland, OR) was used to validate MS/MS-based peptide and protein identifications, and to perform dataset alignment. Peptide identifications established at lower than 90.0% probability by the Scaffold Local FDR algorithm were filtered out. Protein identifications were accepted if they could be established at greater than 95.0% probability and contained at least two identified peptides. Protein probabilities were assigned by the Protein Prophet algorithm[66]. Proteins that contained similar peptides and could not be differentiated based on MS/MS analysis alone were grouped to satisfy the principles of parsimony. Proteins sharing significant peptide evidence were grouped into clusters.

**Immunofluorescence**. Cells grown on coverslips were fixed in 4% formaldehyde/PBS for 20 min at RT, then permeabilized in 0.02% Triton-X/PBS for 10 min. Coverslips were incubated with blocking buffer containing 1% FBS and 2% bovine serum albumin (BSA) for 1 h. Primary antibodies (anti-ARP-T1 (1:100, SAB2103464 or SAB1408334; or 1:200, Progen GP-SH6), anti-acetylated tubulin (1:1000, T6793), anti-rootletin (1:50, sc-67824, Santa-Cruz; 1:200, NBP1-80820, Novus), anti-gamma tubulin (1:500, ab11316, Abcam), anti-EDH4 (1:200, Dr. Plomann's lab), anti-septin 2 (1:200, HPA018481, Sigma-Aldrich), anti-septin 9 (1:200, HPA042564) were diluted in blocking buffer and applied on the cells for 2 h at RT or overnight at 4 °C. Coverslips were next incubated with secondary antibodies (1:500, A11008, A11001, A21467, A11035, A11003, A11060, Invitrogen; 1:100, BA-9500, Vector Laboratories; 1:200, RPN1233V, GE) for 1 h at RT. Nucleus was stained with DAPI or Hoechst for 5 min at RT. Finally, coverslips were mounted onto slides with Dako mounting medium (S3023, Dako Schweitz AG, Baar, Switzerland) and examined with an inverted Zeiss LSM 700 laser scanning confocal microscope equipped with laser diode 405/488/555, SP490/SP555/LP560 emission filters, 2 PMT detectors and Zen2010 software (Zeiss, Feldbach, Switzerland).

Three-dimensional (3D) imaging technique was used to study ciliary imaging. Confocal images were captured at ~0.33–1 µm interval using 63 ×/1.40 oil objectives. Using Image J software, the 3D structure was deconvoluted from the Z-stack, and the length, intensity, and prevalence of cilia were manually traced and measured. Both cell number and proportion of cells exhibiting primary cilia were determined in approximately five representative fields (100–200 µm²) for each experimental condition. The mean cilia prevalence was expressed as the percentage

of ciliated cells. The average ciliary length was expressed in μm. The ciliary rootlet was selected using the freehand tool and the average area was expressed in μm². The average pixel fluorescence intensity of proteins was quantified using the freehand tool to select the area of interest. Twenty to 50 cilia were measured for each experimental condition. Septins signal in the axoneme was quantified as follows: on split channels, the mean gray value of acetylated tubulin was measured using the freehand tool, then with ROI manager, the same area in the septin 2 or septin 9 channel was also measured; percentage of relative intensity = (mean gray value septin/mean gray value acetylated tubulin) × 100.

**RNA extraction and real-time quantitative PCR**. Total RNA was harvested using the Qiagen RNeasy Mini Kit (Qiagen, Hombrechtikon, Switzerland) according to the manufacturer's instructions. RNA quantity and quality were assessed using the NanoDrop ND-1000 Spectophotometer (Wilmington, USA). RNA were reverse-transcribed using the Primescript RT reagent kit (TaKaRa, Saint-Germain-en-Laye, France) in 10 μL reaction on a T-Gradient Thermocycler (Biometra, Biolabo, Châtel-St-Denis, Switzerland). Quantitative PCR (qPCR) was performed on cDNA with Power SYBRGreen PCR Mastermix (Applied Biosystems, CA, USA) on an Applied Biosystems StepOne thermal cycler (Applied Biosystems, Life Technologies, CA, USA). Gene expression was assessed by using ACTRT1 (QT00215642, Qiagen), GLI1 (Fw 5′ AGA GGG TGC CAT GAA GCC AC 3′, Rev 5′ AAG GTC CCT CGT CCA AGC TG 3′, Microsynth), PTCH1 (Fw 5′ GCT ACT TAC TCA TGC TCG CC 3′, Rev 5′ TCC GAT CAA TGA GCA CAG GC 3′, Microsynth) primers. RPL13A (Fw 5′GCA TCC CAC CGC CCT ACG AC 3′, Rev 5′ CTC TTT CCT CTT CTC CTC CA 3′, Microsynth) was used as endogenous control. Relative gene expression is normalized to the control and reported with a log2 scale.

**Proliferation assay**. Proliferation was assessed using the Click-iT EdU Proliferation Assay for Microplates (C10499, Invitrogen). Briefly, cells were plated at 48% confluency in 96 well black microplate (CLS3603, Corning, MERCK), labeled with 10 μM EdU for 24 h, then fixed and clicked according to the manufacturer's instructions. Fluorescence was analyzed with a Mithras LB 940 reader (Berthold technologies, Bad Wildbad, Germany) and MikroWin 2010 software (Berthold technologies) using excitation filter 560 nm and emission filter 590 nm.

**Apoptosis assay**. Cells were harvested in Versene (15040, Gibco, ThermoFisher Scientific), then stained with APC Annexin V (550474, BD Pharmingen, San Jose, CA) and Annexin V Apoptosis Detection Kit I (556547, BD Pharmingen) according to the manufacturer's instructions. Results were collected with a FACS Gallios I (Beckman Coulter, Nyon, Switzerland) and analyzed with FlowJo software (BD Life Sciences) at the Flow Cytometry Facility at Unil.

**Statistics and reproducibility**. We used the one-way ANOVA or the Kruskal–Wallis test, depending on the homogeneity of variances, to compare one independent variable in more than two groups. Linear regression was used to compare the correlation between two parameters.

To compare gene expression quantified by RT-qPCR, for each gene, all samples were normalized to the control, and log-2 transformed fold changes comparison to 0 were performed with a one-sample t test. To compare protein expression, all samples were normalized to the control, and fold changes comparison to 1 were performed with a one-sample t test.

All experiments were performed at least three times. All statistical analyses were performed using Prism GraphPad (v8, La Jolla, CA, USA).

**Reporting summary**. Further information on research design is available in the Nature Research Reporting Summary linked to this article.

## Data availability

The mass spectrometry proteomics data are available on the ProteomeXchange Consortium[67] via the PRIDE[68] partner repository with the dataset identifier PXD016557 and 10.6019/PXD016557.

Original immunoblots and immunofluorescence data are accessible on Zenodo repository[69] with the dataset identifier 10.5281/zenodo.4633301 (all versions: 10.5281/zenodo.3666278). Source data can be found in Supplementary Data 1.

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

## Acknowledgements
The authors thank Prof. Rune Toftgård, Prof. Anthony Oro, and Dr. Sergey Nikolaev for their kind advice. We also acknowledge the Protein Analysis Facility of the University of Lausanne for the mass spectrometry assay. This study was funded by Swiss National Science Foundation (grant 310030-173102), "Fondation Professeur Placide Nicod" and "Fondation Dind Cottier pour la recherche sur la peau".

## Author contributions
HS.P., E.P., G.B., E.C., D.B., and C.P. designed and performed the experiments. M.P., F.MP., P.V., and A.S. provided plasmids, antibodies and the human samples. HS.P., E.P., C.P., and D.H. wrote the manuscript. M.H., C.P., and D.H. supervised the project.

## Competing interests
The authors declare no competing interests.
