## [Peer Review File · Communications Biology]

Reviewers' comments:

Reviewer #1 (Remarks to the Author):

The main claim of this paper is that ARP-T1 is a ciliary protein that directly interacts with the ciliary machinery, is required for ciliogenesis, and whose mutation causes a new ciliopathy (BDCS) that is associated with hair follicle defects and BCC. Although the data in the paper are intriguing, not all of the claims are strongly supported by the data, which is incomplete in various ways. In particular the mechanism by which ARP-T1 affects ciliogenesis is apparently via disorganization of the actin cytoskeleton, but there is no data with either the KD or patient samples that show this is the case. If the authors can provide additional experimental evidence linking the patient mutation/truncation to defective ciliogenesis (or actin cytoskeletal organization) I am willing to review a revised version of the manuscript.

Specific comments:

Can the authors show (or at least comment on) the functionality of the truncated protein resulting from the frame shift mutation in ARL-T1 identified in BDCS patients? The paper would be more convincing if a rescue experiment was performed to test the effects of ARP-T1 mutant truncation on ciliogenesis. For example, the ARP-T1 with the mutated portion present in patient samples could be expressed in the ARP-T1 KD cells: does this rescue the ciliogenesis defect or do cilia remain shortened? These data would convince me of the role of ARL-T1 in ciliogenesis. Although the cilia data using the patient samples is convincing, the cell line data is less so. I am wondering how truncation of ARP-T1 directly affects ciliogenesis, or is it a consequence of function at mid-body or defects in cell cycle, for example. The overall structure and organization of the actin cytoskeleton should also be examined in all of these experiments, given ARP-T1's previously described roles.

Note for text (PAGE 3)- I disagree with the statement that IFT88 is considered both a differentiation protein and a ciliary protein. Please provide references supporting the statement that Ift88 is involved in differentiation or considered a differentiation marker, protein, etc.

Figure 1 shows that ARP-T1 expression increases upon differentiation in cultured human keratinocytes. Does ciliogenesis also increase? (this has been shown for mouse keratinocytes, but I am unaware of data for human cells).

Figure 2- the authors show that SAG treatment induces ARP-T1 expression. What happens to ARP-T1 localization at cilia/basal body/transition zone (or elsewhere) upon SAG treatment?

Figure 3. The authors show that ARP-T1 interacts with acetylated tubulin and Septins, and this data forms the basis of the claim that ARP-T1 is a ciliary protein that interacts with the ciliary machinery. However, these components all have diverse, non-ciliary localizations and functions throughout the cell. Further, ARP-T1 is mostly an actin associated protein. Taken together, I don't think one can make the claim that ARP-T1 is interacting with the ciliary machinery (which I would define as IFTs, kinesins or other regulatory proteins at the axoneme). In addition, the localization data is really not very convincing, as they seem to show different types of localization at what I presume is the diffusion barrier/transition zone at the base of the cilium. If ARP-T1 is localized with septins at the diffusion barrier at the base of the cilium, how does its loss cause shortened cilia? Are Septins mislocalized at the base of the cilium in the ARP-T1 deficient cells? Maybe there are cell cycle defects (given the mid-body localization), and this indirectly causes the shortened cilium?

At various points in the paper the authors state that ARL-T1 causes shortened cilia due to defect in actin cytoskeleton, but I can't find the data. Do ARL-T1 KD cells have disorganized actin? Is this seen in the patient samples? Why is this claim made? Does pharmacological disruption of the actin cytoskeleton disrupt ARL-T1 localization? This is an essential missing component- the authors should try using immunofluorescence to show the structure and organization of the actin cytoskeleton in cells/patient samples where ARL-T1 function is disrupted.

Reviewer #2 (Remarks to the Author):

The authors show that activation of the HH pathway increases ACTRT1 mRNA in a non-canonical manner. They show localization of ARP-T1 to the basal body and midbody during cytokinesis. Their data suggests PKC phosphorylates and stabilizes ARP-T1, and further show that patient-derived BCC with mutant ARP-T1 have a lower expression of this protein and shorter cilia.

The MS analysis demonstrated various interactions with proteins involved in ciliogenesis, and the discussion of these findings raised many hypotheses for function in this context.

This study demonstrates original data that is of interest to the field of primary cilia biology and furthers our knowledge on how it relates to tumorigenesis. Strengths include use of in-vitro models and mass spectrometry paired together with patient-derived samples. While there is solid evidence for various interactions and localization at the cilium and centrioles, the consequence of shortened cilia is still very descriptive. It is unclear how the shortened cilia relates to activation of the HH pathway as a key driver in BCC pathogenesis. The manuscript would be strengthened by better stating how these findings fit into our knowledge of BCC pathogenesis.

Reviewer #3 (Remarks to the Author):

Park et al demonstrated that ARPT1 is a novel ciliary component and loss of ARPT1 inhibits ciliogenesis in keratinocytes. Although the role of primary cilia in skin cancer has been investigated, the authors showed that primary cilia regulated by ARPT1 is involved in pathogenesis of inherited BCC. Several additional studies will help to improve the story.

Major comments

- 1) In Figure 4, the authors showed that knock down of ARPT1 decrease cilium length RPE1 and HaCat cells. primary cilium is critical for various cell signaling. How about the function of primary cilia in ARPT1 K/D cells? In addition, the western blots in Fig 4j and 4n is not clear (use at least two different siRNAs to avoid off-target effect).
- 2) In Figure 2, the authors have to show expression level of the proteins.
- 3) the author showed ARPT1 interacting proteins in HeLa and RPE cells but did not in NHEK cells (Data not shown). But I think the result from NHEK cells is more important than from HeLa cells to support the title.
- 4) There are so many typo errors in text and figures such as HH vs SHH, hour vs h, CO2 vs CO2, and space. The authors have to revise the manuscript carefully.

Manuscript # COMMSBIO-20-0266-T

REVISED VERSION

RESPONSES TO THE REVIEWERS

Reviewer #1 (Remarks to the Author):

The main claim of this paper is that ARP-T1 is a ciliary protein that directly interacts with the ciliary machinery, is required for ciliogenesis, and whose mutation causes a new ciliopathy (BDCS) that is associated with hair follicle defects and BCC. Although the data in the paper are intriguing, not all of the claims are strongly supported by the data, which is incomplete in various ways. In particular the mechanism by which ARP-T1 affects ciliogenesis is apparently via disorganization of the actin cytoskeleton, but there is no data with either the KD or patient samples that show this is the case. If the authors can provide additional experimental evidence linking the patient mutation/truncation to defective ciliogenesis (or actin cytoskeletal organization) I am willing to review a revised version of the manuscript.

Answer:

We thank the reviewer for his/her comments. We hope that we fully answered his/her concerns in the revised manuscript and the following answers.

Specific comments:

Can the authors show (or at least comment on) the functionality of the truncated protein resulting from the frame shift mutation in ARP-T1 identified in BDCS patients?

Answer:

In our previous publication, Bal *et al*, 2017, we described the truncated ARP-T1 as a loss of function. We added this comment in the introduction (line 37).

“The insertion mutation ACTRT1 547_548InsA, creating a shift of the reading frame, results in a truncated protein of 198 amino acids **leading to a loss of function of the protein**. ARP-T1 was also found depleted in families with germline mutations of non-coding sequences surrounding *ACTRT1* postulated to belong to enhancers transcribed as non-coding RNAs. As these mutations segregate with the disease, ARP-T1 can be considered as a tumor suppressor in BDCS.”

The paper would be more convincing if a rescue experiment was performed to test the effects of ARP-T1 mutant truncation on ciliogenesis. For example, the ARP-T1 with the mutated portion present in patient samples could be expressed in the ARP-T1 KD cells: does this rescue the ciliogenesis defect or do cilia remain shortened? These data would convince me of the role of ARP-T1 in ciliogenesis. Although the cilia data using the patient

samples is convincing, the cell line data is less so. I am wondering how truncation of ARP-T1 directly affects ciliogenesis, or is it a consequence of function at mid-body or defects in cell cycle, for example. The overall structure and organization of the actin cytoskeleton should also be examined in all of these experiments, given ARP-T1's previously described roles.

Answer:

To confirm the role of ARP-T1 in ciliogenesis, we overexpressed ARP-T1 WT and mutant in hTERT-RPE1 (revised Fig. 4g,h and Extended data Fig. 4a) and in HaCaT cells (data not shown). The ciliary length is the same comparing ARP-T1 WT overexpression and vector condition, but is shortened with the ARP-T1 mutant. These results correlate with the ones that we observed in patients' samples (Fig 4a).

We performed the requested expression of ARP-T1 mutant in hTERT-RPE1 *ACTRT1* KD cells. We first mutated the ARP-T1 mutant plasmid to remove shRNA1 and shRNA2 targets (MshR), but producing the same protein (same AA with different nucleotides). After transduction in the *ACTRT1* KD hTERT-RPE1 cells, and verification of the good expression of ARP-T1 MshR, we measured the ciliary length. ARP-T1 MshR reduced the ciliary length in control cells, and did not induce further reduction in KD cells (revised Fig. 4l and Extended data Fig. 6a). These results confirmed that ARP-T1 loss of function and *ACTRT1* silencing both lead to identical reduction of the ciliary length.

Revised Fig. 4 from g to o.

New extended data Fig. 4a.

New extended data Fig. 6a.

To detect a role of ARP-T1 in the midbody, we used the strategy inspired from Pejškova et al. (*J Cell Biol* 2020). They described that KIF14, a protein involved in ciliogenesis, was also located to the midbody and they analyzed the impact of the protein on cell division. We analyzed *ACTRT1* KD hTERT-RPE1 cells proliferation and apoptosis. First, *ACTRT1* silencing does not affect the ability of the cells to grow, as shown by EdU incorporation assay (revised Fig. 5d). Second, *ACTRT1* decrease has no impact on apoptosis, assessed by Annexin V/ propidium iodide analysis (revised Fig. 5e, Results lines 246 to 252, and Discussion lines 353 to 356). Despite the midbody localization, and as already reported for KIF14, ARP-T1 has no effect on cell division.

Revised Fig. 5d,e.

Note for text (PAGE 3)- I disagree with the statement that IFT88 is considered both a differentiation protein and a ciliary protein. Please provide references supporting the statement that Ift88 is involved in differentiation or considered a differentiation marker, protein, etc. Figure 1 shows that ARP-T1 expression increases upon differentiation in cultured human keratinocytes. Does ciliogenesis also increase? (this has been shown for mouse keratinocytes, but I am unaware of data for human cells).

Answer:

We apologize to the reviewer for not being clear enough. We hope that the following information will convince him/her. Differentiation is characterized by ciliogenesis in RPE cells (Katoh *et al.*, *Mol Biol Cell* 2017; Hazim *et al.*, *Exp Eye Res* 2018) and keratinocytes (Bae *et al.*, *Sci Rep* 2019). Pugacheva *et al.* (*Cell* 2007) showed that IFT88 is essential for cilia formation in hTERT-RPE1 cells, and that IFT88 expression decreases following cilia disassembly. Therefore, we used IFT88 expression as a differentiation marker. We added these comments and references to the results section (lines 81 to 92) and our results on increased number of ciliated cells in the revised extended data Fig. 1a.

“These cells were differentiated under serum starvation for 35 days or 48 h respectively. During RPE differentiation, ciliogenesis is known to increase^{14,20,21} and IFT88 is a well-characterized protein involved in RPE ciliogenesis^{22,23}. We assessed RPE differentiation monitoring the expression of the ciliary protein IFT88 (Fig. 1f,h), and increased number of ciliated cells (Extended data Fig. 1a).”

Figure 2- the authors show that SAG treatment induces ARP-T1 expression. What happens to ARP-T1 localization at cilia/basal body/transition zone (or elsewhere) upon SAG treatment?

Answer:

We treated hTERT-RPE1 cells with SAG or purmorphamine and analyzed by the localization of ARP-T1 by immuno-fluorescence. We found that ARP-T1 remains at the cilia basal body upon SMO activation, according to our double-staining with rootletin and septin 2. These results were added in the revised Fig. 4m and in the results section (lines 220 to 226).

Revised Fig. 4m.

Figure 3. The authors show that ARP-T1 interacts with acetylated tubulin and Septins, and this data forms the basis of the claim that ARP-T1 is a ciliary protein that interacts with the ciliary machinery. However, these components all have diverse, non-ciliary localizations and functions throughout the cell. Further, ARP-T1 is mostly an actin associated protein. Taken together, I don't think one can make the claim that ARP-T1 is interacting with the ciliary machinery (which I would define as IFTs, kinesins or other regulatory proteins at the axoneme). In addition, the localization data is really not very convincing, as they seem to show different types of localization at what I presume is the diffusion barrier/transition zone at the base of the cilium. If ARP-T1 is localized with septins at the diffusion barrier at the base of the cilium, how does its loss cause shortened cilia? Are Septins mislocalized at the base of the cilium in the ARP-T1 deficient cells? Maybe there are cell cycle defects (given the mid-body localization), and this indirectly causes the shortened cilium?

Answer:

In the current study, the term “ciliary machinery” refers to all the proteins important for ciliogenesis and which are located from the basal body to the tip of the axoneme. We rephrased the title (ARP-T1 is a ciliogenesis protein, line1) and the text (we referred to a basal body localization instead of a ciliary localization in the revised manuscript). We detected ARP-T1 in three major compartments of the cells: cytoplasm, basal body and midbody, depending on the cell cycle stages.

We performed experiments showing that remaining ARP-T1, in *ACTRT1* KD cells, still co-localizes with rootletin, septins 2 and 9 (additional extended data Fig. 6b, lines 226 and 227). We also showed that septin 9 localized to the axoneme, septin filaments and the base of the PC in control and *ACTRT1* KD cells under differentiation. Therefore, ARP-T1 decrease does not affect neither its interactions with rootletin and septins nor septin 9 localization. Unexpectedly, we observed that septin 2, normally localized in the axoneme under serum-starvation, is no longer

there in *ACTRT1* KD cells. We were able to detect septin 2 in its other locations, septin filaments and the base of the PC. These results suggest that the ciliogenesis defect observed in *ACTRT1* deficient cells could be due to a default in the diffusion barrier mediated by septin 2. We added these results in the revised Fig 4n, in the results (lines 227 to 233) and the discussion (lines 339 to 349). We are currently investigating the link between septin 2 and ARP-T1 in cilia formation. This part requires many more experiments before giving a complete mechanism, notably using SMO in/out assay (Kukic *et al.*, *Cilia* 2016) and FRAP experiments (Ghossoub *et al.*, *J Cell Sci* 2013). These results will be part of a forthcoming publication.

New extended data Fig. 6b.

Revised Fig. 4n.

At various points in the paper the authors state that ARP-T1 causes shortened cilia due to a defect in the actin cytoskeleton, but I can't find the data. Do ARP-T1 KD cells have disorganized actin? Is this seen in the patient samples? Why is this claim made? Does pharmacological disruption of the actin cytoskeleton disrupt ARP-T1 localization? This is an essential missing component- the authors should try using immunofluorescence to show the structure and organization of the actin cytoskeleton in cells/patient samples where ARP-T1 function is disrupted.

Answer:

We agree with the reviewer, and we apologize for the miswriting. In our initial manuscript version, we suggested that ARP-T1-related shortening of cilia was due to a defect in actin cytoskeleton network, as ARP-T1 was found to interact with several actin cytoskeleton proteins

(cortactin and septins for instance), but we did not clearly show their role in deficient ciliogenesis. Thus, we performed different experiments to better understand how ARP-T1 affects ciliogenesis.

We first treated hTERT-RPE1 cells with cytochalasin D. Using the same protocol as described by Gonçalves *et al.* (*J Cell Biol* 2020), we induced ciliogenesis in proliferating cells. The number of ciliated cells and the cilia length were increased when the cells were treated with cytochalasin D, with or without ARP-T1 (revised Fig. 4o, Results lines 234 to 240, and Discussion lines 335 to 338). These results indicate that the actin cytoskeletal network is not deficient in these cells and is unlikely to be responsible for the shortening of the ciliary length.

As we noted in our response to the previous question, we found that septin 2 is absent from the axoneme of *ACTRT1* KD cells, and that we are currently investigating the role of septin 2 and ARP-T1 interaction in cilia formation. Our current hypothesis is that ARP-T1 is important for septin 2 localization and full length of the axoneme.

Revised Fig. 4o.

Reviewer #2 (Remarks to the Author):

The authors show that activation of the HH pathway increases *ACTRT1* mRNA in a non-canonical manner. They show localization of ARP-T1 to the basal body and midbody during cytokinesis. Their data suggest PKC phosphorylates and stabilizes ARP-T1, and further show that patient-derived BCC with mutant ARP-T1 have a lower expression of this protein and shorter cilia.

The MS analysis demonstrated various interactions with proteins involved in ciliogenesis, and the discussion of these findings raised many hypotheses for function in this context.

This study demonstrates original data that is of interest to the field of primary cilia biology and furthers our knowledge on how it relates to tumorigenesis. Strengths include use of in-vitro models and mass spectrometry paired together with patient-derived samples. While there is solid evidence for various interactions and localization at the cilium and centrioles, the consequence of shortened cilia is still very descriptive. It is unclear how the shortened

cilia relates to activation of the HH pathway as a key driver in BCC pathogenesis. The manuscript would be strengthened by better stating how these findings fit into our knowledge of BCC pathogenesis.

Answer:

We thank the reviewer of his/her positive comments.

BDCS is a novel ciliopathy, a derivative of BCC, where ARP-T1 plays a critical role, regulated by non-canonical HH pathway to prevent BCC pathogenesis. The activation of a non-canonical HH pathway may explain the apparent paradox that we observed shortened PC both in BCC from BDCS and in *ACTRT1* KD cells. While loss of PC is observed in diverse cancers where it correlates with worse prognosis, their retention is paramount for BCC development. Moreover, resistant BCCs are no longer sensitive to vismodegib and lose PC (Zhao *et al.*, *Cancer Discov* 2017; Kuonen *et al.*, *J Invest Dermatol.* 2019). Glaessl *et al.*, (*Dermatol Surg* 2000) described that BDCS BCCs behave like aggressive BCCs and are prone to relapse. This suggest that BDCS is a model for the studying of vismodegib resistance in BCC. We added this part to the discussion (lines 410 to 412).

Functions of the PC, including activation of the HH signaling, are dependent on the ciliary length and on proteins involved in ciliogenesis, such as proteins located to the basal body (Bangs *et al.*, *Cold Spring Harb Perspect Biol* 2017). In addition to this, ARP-T1 was reported to regulate the HH pathway (Bal *et al.*, *Nat. Med* 2017). We investigated a potential defect in PC function, in *ACTRT1* KD hTERT-RPE1 cells, by analyzing the expression of two direct targets of the HH signaling pathway. *GLI1* and *PTCH1* mRNA expression tends to increase but not significantly in *ACTRT1* KD cells (additional extended data Fig. 6c,d), suggesting that our silencing might not be sufficient to correctly activate the HH signaling, or that another hit might be needed for a proper activation.

We added these results to the discussion and deepened the discussion on this topic (lines 382 to 402).

New extended data Fig. 6.

Reviewer #3 (Remarks to the Author):

Park et al demonstrated that ARPT1 is a novel ciliary component and loss of ARPT1 inhibits ciliogenesis in keratinocytes. Although the role of primary cilia in skin cancer has been investigated, the authors showed that primary cilia regulated by ARPT1 is involved in pathogenesis of inherited BCC. Several additional studies will help to improve the story.

Major comments

1) in Figure 4, the authors showed that knock down of ARPT1 decrease cilium length RPE1 and HaCat cells. primary cilium is critical for various cell signaling. How about the function of primary cilia in ARPT1 K/D cells? In addition, the western blots in Fig 4j and 4n is not clear (use at least two different siRNAs to avoid off-target effect).

Answer:

Functions of the PC, including activation of the HH signaling, are dependent on the ciliary length and on proteins involved in ciliogenesis, such as proteins located to the basal body (Bangs *et al.*, *Cold Spring Harb Perspect Biol* 2017). In addition to this, ARP-T1 was reported to regulate the HH pathway (Bal *et al.*, *Nat. Med* 2017). We investigated a potential defect in PC function, in *ACTRT1* KD hTERT-RPE1 cells, by analyzing the expression of two direct targets of the HH signaling pathway. *GLI1* and *PTCH1* mRNA expression tends to increase but not significantly in *ACTRT1* KD cells (additional extended data Fig. 6c,d), suggesting that our silencing might not be sufficient to correctly activate HH signaling, or that another hit might be needed for a proper activation.

We added these results to the discussion and deepened the discussion on this topic (lines 382 to 402).

We are currently studying in the same cells the impact of SMO activation with SAG and purmorphamine. These results will be part of a forthcoming publication.

New extended data Fig. 6.

We apologize for the lack of a second shRNA in HaCaT cells. Since our first submission, we found a second shRNA (shRNA3) which successfully decreased *ACTRT1* expression (additional extended data Fig5, Results lines 204 and 205). This second shRNA confirms the results we previously obtain with the first shRNA in HaCaT cells.

New extended data Fig. 5.

2) In Figure 2, the authors have to show expression level of the proteins.

Answer:

We did not analyze ARP-T1 expression for all the treatments, as its expression is quite difficult to detect, and we preferred to demonstrate on the impact of *ACTRT1* KD or mutant on the ciliary length, rather than to uncover the full mechanism of ARP-T1 regulation.

We focused on the experiments modulating the HH signaling pathway. We performed protein analyses for ARP-T1, GLI1 and Patched. On the one hand, we were not able to detect GLI1 and Patched at the protein level in our cells. On the other hand, we correctly detected ARP-T1 expression under the HH signaling pathway modulation, we added the results in the revised Fig. 2 (Results lines 103, 112 and 113, Discussion lines 277, 371 to 374).

Revised Fig. 2 from a to e.

3) the author showed ARPT1 interacting proteins in HeLa and RPE cells but did not in NHEK cells (Data not shown). But I think the result from NHEK cells is more important than from HeLa cells to support the title.

Answer:

We agree with the reviewer comment and we added the results from the NHEK cells in the table 1 (line 151). They fully support the results we submitted in our first manuscript.

4) There are so many typo errors in text and figures such as HH vs SHH, hour vs h, CO2 vs CO2, and space. The authors have to revise the manuscript carefully.

Answer:

We thank the reviewer for the notice. We carefully checked the revised manuscript and hope to have corrected all typo errors.

REVIEWERS' COMMENTS:

Reviewer #3 (Remarks to the Author):

the authors completely cleared the previous queries.

Editor:

It is my opinion that the authors did a good effort to address the concerns and suggestions of the reviewers. However, some of the experimental data require further quantification or better examples before I can support publication in Communications biology:

1. In Figure 2b: the western quantification of ARP-T1 levels does not seem to match the blot. Could there be any mistake in the quantification of the signal? If not, do the authors have a better, more representative blot they can use instead?
2. In Figure 4o: the units on the Y-axis are decimal but it is labeled as %. Is this a mistake or are the ciliogenesis values that low (i.e. 2 ciliated cells/1000 cells in DMSO!) ?
3. Figures 4m and 4n represent very important results for the conclusions of this manuscript as the authors show that ACTRT1 KD leads to the abnormal recruitment of septin 2 to the ciliary axoneme, which might be related to the shorter cilia seen in ACTRT1-associated BDCS samples. Therefore: I) it seems important to clearly show the separated channels (maybe even in greyscale) for the reader to see this effect more clearly; II) the authors should try to quantify either the signal of both septin 2 and 9 that colocalizes with the ciliary axoneme in control and ACTRT1 KD cells.
4. In Supplementary Figure 5A: The ACTRT1 KD3 in HaCaT cells only has 3 cells, one of which is enormous. Is this the most representative example the authors have? If so, it suggests the occurrence of defects in cell division...
5. Are the Supplementary Figure 2 and 3 the same data but different examples? If so, can the authors consider one Supplementary Fig 2A and the other Supplementary 2B, stating that these are additional examples?

In addition, the authors should carefully correct the following mistakes/ improve the text of their manuscript to simplify and better transmit their findings to the average reader:

I strongly suggest simplifying the title by removing "a ciliogenesis protein". In fact, my suggestion would be to rephrase it to something along these lines:

"ARP-T1-associated Bazex-Dupré-Christol Syndrome is an inherited basal cell cancer with ciliary defects characteristic of ciliopathies"

Line 20 – in "interactome (PXD016557) involved in ciliogenesis" edit to "interactome (PXD016557) that includes proteins involved in ciliogenesis"

Line 21 – Replace "Consequently" by "In agreement," or simply remove it.

Line 22 – Remove "in G0" as cells were not co-stained with G0 specific cell cycle markers. Or at most replace it with "in interphase"

Line 22-25 – I strongly suggest editing: "Tissue samples from BDCS patients show reduced ciliary length with significant correlations of ARP-T1 expression levels, confirmed by ACTRT1 knock down. We report that BDCS is a novel ciliopathy and the first case of a skin cancer ciliopathy, where ARP-T1 plays a critical role to prevent pathogenesis." following these lines:

"Tissue samples from patients with ACTRT1-associated BDCS have reduced ciliary length. The severity of the shortened cilia significantly correlates with the ARP-T1 levels, which was further validated by ACTRT1 knock down in culture cells. Thus, we propose that ARP-T1 participates in the regulation of cilia length and that ACTRT1-associated BDCS is a case of a skin cancer with ciliopathy characteristics."

Line 36 – Improve the following sentence: “The insertion mutation ACTRT1 547_548InsA, creating a shift of the reading frame, results in a truncated protein of 198 amino acids leading to a loss of function of the protein.”

i.e. “The insertion mutation ACTRT1 547_548InsA creates a shift in the reading frame that results in a non-functional truncated protein of 198 amino acids.”

Line 45 – Replace “...either motile as in sperm propel or nonmotile (primary) acting as sensory antenna, receiving signals...” by “...either motile, such as the sperm flagella, or nonmotile, such as the primary cilium that acts as a sensory antenna, receiving signals...”

Line 57-58- Replace “PC assembly is a complex process emanating from the mother centriole with its appendages and differentiating upon cell cycle exit” by:

“PC assembly is a complex process that involves the attachment of the mother centriole with its appendages to the plasma membrane in G1 or upon cell cycle exit.”

Line 71-72- The authors overstate: “Here, we demonstrate that ARP-T1 is a basal body protein and is involved in ciliogenesis by interacting with the ciliary machinery.” It should be toned down to something like this “Here, we show that ARP-T1 localizes to the basal body, interacts with several components of the ciliary machinery and contributes to cilium extension.”

Line 73-74: The authors overstate: “...give rise to the abnormally shortened cilia, and this may be caused by a disordered diffusion barrier.” ...as the integrity of the transition zone/diffusion barrier was not assessed in this study. It should be toned down to something like this “...give rise to the abnormally shortened cilia, and this may be caused by the displacement of septin 2.”

Line 77: Replace “We report that BDCS is the first ciliopathy of epidermal development and cancer.” by “We report for the first time the presence of cilia defects in ACTRT1-associated BDCS epidermal development and cancer, and propose that this pathology should be considered a ciliopathy.”

Line 180: Replace “ARP-T1 associates with a ciliopathy in BDCS.” The authors should rephrase this title to: “ARP-T1-associated BDCS tissues have shorter cilia.”

Line 181- 182: “Based on the localization of ARP-T1 to the basal body and the implication of septin 2 in ciliary length control, we...”. The following clarification should help the reader to better understand the author's rationale for analyzing ciliary length: “Based on the localization of ARP-T1 to the basal body and its interaction with septin 2, that is involved in ciliary length control, we...”

Line 193-195: The authors should rephrase this sentence: “We concluded that BDCS, inherited BCC, unlike sporadic BCC, is a novel ciliopathy and the role of ARP-T1 as a part of cilia is to prevent the disease.”

to “These results suggest that ARP-T1 is important for cilia length and that ARP-T1-associated BDCS might originate from ciliary defects.”

Line 220 – 221: “ARP-T1 favors for septin 2 localization in the axoneme, without affecting the actin cellular network.” This sentence is not correct: the first half refers to the protein function while the second half refers to ARP-T1 loss/mutation/depletion. I suggest rephrasing to “ARP-T1 loss of function disrupts septin 2 localization to cilia without affecting the actin cellular network.”

Line 242: I suggest improving this title from “ARP-T1 is located in the midbody in dividing cells.” to “ARP-T1 also localizes to the midbody in dividing cells”.

Line 257-258: I suggest improving this sentence: “Both midbodies and PC contain acetylated tubulin, many proteins in the midbody can also be found at the base of the cilium in the centrioles (Fig. 5f-i).” to:

“Both midbodies and PC contain acetylated tubulin and several proteins at the midbody have been found to relocate to the base of assembling cilia to participate in ciliogenesis and in cilia-mediated signaling (Fig. 5f-i).”.

Reference "29" seems to be incorporated into the manuscript by mistake as it's completely unrelated to every sentence that it's associated with in the manuscript.

In lines 256 to 258 – References 28 to 30 are incorrectly used. Appropriate references must be used here. I assume that these mistakes were unintentional and I did not find any other cases of incorrect references but I strongly advise the authors to reconfirm all the references in their manuscript before submitting their final version!

Line 266 – Rephrase: "Here, we report that ARP-T1 is a ciliogenesis protein located at the basal body of PC." To "Here, we report that ARP-T1 localizes to the basal body of PC where its involved in the regulation of the extension of the ciliary axoneme."

Lines 327 to 338 need to be greatly clarified to make a more logical and easier to follow flow for the reader. If I understood the ideas in the text correctly then it might help to:

-take the sentence "Despite all these clues,... in the shortened cilia observed in silenced ACTRT1 cells." (line335-338)

-and move it to line327 after "...ciliogenesis." and before "Similar to ARP-T1,..."

Line 327: "ciliary resorption" should be replaced by "shorter cilia or faster ciliary resorption".

Line 339: Replace "Diffusion barrier..." by "The transition zone or diffusion barrier..."

Line 344-347: Replace to improve flow and avoid overstating: "Such a situation was different for septin 9, whose location remained unchanged, i.e. in the axoneme, septin filaments and the base of the PC. These results suggest that the ciliogenesis defect observed in ACTRT1 deficient cells could be due to a default in the diffusion barrier mediated by septin 2." by:

"In contrast, the localization of septin 9 at the axoneme and base of cilia, and in septin filaments remained unchanged. These results suggest that the ciliogenesis defect observed in ACTRT1 deficient cells could be related to the mislocalization of septin 2, as it is required for normal axoneme extension (ref24)."

Line 347-349: "We are currently investigating the link between septin 2 and ARP-T1 in cilia formation, notably using SMO in/out assay and FRAP experiments. These results will be part of a forthcoming publication." should be rephrased to a more standard form such as "Future experimental work will be needed to further investigate the link between septin 2 and ARP-T1 in cilia formation."

Line 357: Replace "leads" by "contributes to" or "participates in"

Line 358: Replace "in" by "of the"

Line 388: I suggest replacing "or that another hit might be needed for a proper activation." by "or that other elements might compensate to prevent a stronger activation of the pathway."

Line 403: Replace "Our results show that ARP-T1 acts as a direct or indirect actor in a non-canonical HH pathway" by "Our results suggest that ARP-T1 is directly or indirectly involved in a non-canonical HH pathway".

Line 405: Remove "...the centrosome, the centrioles and..." to simplify the extremely long sentence.

Line 406: Replace "Above all, our studies shed light on how ciliogenesis is controlled in carcinogenesis by ARP-T1" by "This study also sheds light on how ARP-T1-associated ciliary defects might contribute to carcinogenesis"...

Reviewer #3 (Remarks to the Author):

the authors completely cleared the previous queries.

Answer:

We are very thankful to the reviewer for his/her approval.

Editor:

It is my opinion that the authors did a good effort to address the concerns and suggestions of the reviewers. However, some of the experimental data require further quantification or better examples before I can support publication in Communications biology:

1. In Figure 2b: the western quantification of ARP-T1 levels does not seem to match the blot. Could there be any mistake in the quantification of the signal? If not, do the authors have a better, more representative blot they can use instead?

Answer:

We are thank the editor for her positive comments.

We replaced the representative blot in Figure 2b. The previous blot and the new one are on the same membranes (Fig.2b 20201112_1048 P41 ARP.T1 mk.png and Fig.2b 20201116_1536 P41 Actin mk.png) and both were used for the quantification. We hope this second blot is more representative.

Fig. 2b. ARP-T1 expression in SAG-treated hTERT-RPE1 under differentiated condition.

2. In Figure 4o: the units on the Y-axis are decimal but it is labeled as %. Is this a mistake or are the ciliogenesis values that low (i.e. 2 ciliated cells/1000 cells in DMSO!) ?

Answer:

It was indeed a mistake, we corrected the figure.

Fig. 4o. % of ciliated cells in control (Cont.) and ACTRT1 KD (KD1 and KD2) hTERT-RPE1 cells after treatment with cytochalasin D (CytD) or vehicle (DMSO).

3. Figures 4m and 4n represent very important results for the conclusions of this manuscript as the authors show that ACTRT1 KD leads to the abnormal recruitment of septin 2 to the ciliary axoneme, which might be related to the shorter cilia seen in ACTRT1-associated BDCS samples. Therefore: I) it seems important to clearly show the separated channels (maybe even in greyscale) for the reader to see this effect more clearly; II) the authors should try to quantify either the signal of both septin 2 and 9 that colocalizes with the ciliary axoneme in control and ACTRT1 KD cells.

Answer:

We added the images of the separated channels as greyscale and the quantification of the signal intensity of septin 2 and septin 9 in the axoneme in the new Supplementary Figure 6e-h. Using ImageJ and freehand tool, we measured the mean grey value of acetylated tubulin on split channels, then using ROI manager, we also measured the same area in the septin 2 or septin 9 channel. We report here a percentage of relative intensity:

$(\text{mean grey value septin} / \text{mean grey value acetylated tubulin}) \times 100$.

We confirm that the septin 9 signal is similar in the axoneme of control and ACTRT1 KD cells, and that septin 2 signal is decreased in the axoneme of ACTRT1-deficient cells compared to the control.

Supplementary Fig. 6e-h. Individual staining of acetylated tubulin and septin 9 (e) or septin 2 (f) in control and ACTRT1-deficient (KD1 and KD2) hTERT-RPE1 cells under differentiated condition. The white arrows show the primary cilia localization. Quantification of septin 9 (g) and septin 2 (h) signal in the axoneme. Results are presented as Tukey box-plot. Black circles represent outliers (N=20-35).

4. In Supplementary Figure 5A: The ACTRT1 KD3 in HaCaT cells only has 3 cells, one of which is enormous. Is this the most representative example the authors have? If so, it suggests the occurrence of defects in cell division...

Answer:

We thank the editor for her comment. It happens very often to find big nuclei in 7 days-differentiated HaCaT cells. We have examples that are more representative and we replaced the images of KD3 cells. Here the nuclei in KD3 HaCaT are similar to the ones in control and KD1 cells in the new Supplementary Figure 5a.

Supplementary Fig. 5a. Immunofluorescence stainings of acetylated-tubulin (green) and rootletin (red) in 7 days-differentiated control and ACTRT1 KD HaCaT cells. Nuclei are stained with DAPI (blue). Scale bar, 10 μ m.

5. Are the Supplementary Figure 2 and 3 the same data but different examples? If so, can the authors consider one Supplementary Fig 2A and the other Supplementary 2B, stating that these are additional examples?

Answer:

Supplementary Figures 2 and 3 are the same data as Figure 4a, but showing the individual staining. We corrected the text to make it clearer:

“First, we observed the ciliary structures by staining with rootletin and acetylated-tubulin (Fig. 4a, top, **individual stainings are shown in Supplementary Fig. 2**) and confirmed the co-localization of ARP-T1 and rootletin (Fig. 4a bottom, **individual stainings are shown in Supplementary Fig. 3**).”

In addition, the authors should carefully correct the following mistakes/ improve the text of their manuscript to simplify and better transmit their findings to the average reader:

I strongly suggest simplifying the title by removing “a ciliogenesis protein”. In fact, my suggestion would be to rephrase it to something along these lines:

“ARP-T1-associated Bazex-Dupré-Christol Syndrome is an inherited basal cell cancer with ciliary defects characteristic of ciliopathies”

Answer:

We thank the editor for her suggestion and we modified the title.

According to the guidelines, we replaced “Extended data” by “Supplementary”, and the abbreviation “PC” by “primary cilium” or “primary cilia”.

We also wrote RNA as not italics.

We then corrected the text according to her following suggestions.

Line 20 – in “interactome (PXD016557) involved in ciliogenesis” edit to “interactome (PXD016557) that includes proteins involved in ciliogenesis”

Correction line 20-21

Line 21 – Replace “Consequently” by “In agreement,” or simply remove it.

Correction line 21

Line 22 – Remove “in G0” as cells were not co-stained with G0 specific cell cycle markers. Or at most replace it with “in interphase”

Correction line 22

Line 22-25 – I strongly suggest editing: “Tissue samples from BDCS patients show reduced ciliary length with significant correlations of ARP-T1 expression levels, confirmed by ACTRT1 knock down. We report that BDCS is a novel ciliopathy and the first case of a skin cancer ciliopathy, where ARP-T1 plays a critical role to prevent pathogenesis.” following these lines:

“Tissue samples from patients with ACTRT1-associated BDCS have reduced ciliary length. The severity of the shortened cilia significantly correlates with the ARP-T1 levels, which was further validated by ACTRT1 knock down in culture cells. Thus, we propose that ARP-T1 participates in the regulation of cilia length and that ACTRT1-associated BDCS is a case of a skin cancer with ciliopathy characteristics.”

We used “**ARP-T1-associated BDCS**” instead of “ACTRT1-associated BDCS” in the whole manuscript.

Line 36 – Improve the following sentence: “The insertion mutation ACTRT1 547_548InsA, creating a shift of the reading frame, results in a truncated protein of 198 amino acids leading to a loss of function of the protein.”

i.e. “The insertion mutation ACTRT1 547_548InsA creates a shift in the reading frame that results in a non-functional truncated protein of 198 amino acids.

Correction new line 38

Line 45 – Replace “...either motile as in sperm propel or nonmotile (primary) acting as sensory antenna, receiving signals...” by “...either motile, such as the sperm flagella, or nonmotile, such as the primary cilium that acts as a sensory antenna, receiving signals...”

Correction new line 47

Line 57-58- Replace “PC assembly is a complex process emanating from the mother centriole with its appendages and differentiating upon cell cycle exit” by: “PC assembly is a complex process that involves the attachment of the mother centriole with its appendages to the plasma membrane in G1 or upon cell cycle exit.”

Correction new line 59

Line 71-72- The authors overstate: “Here, we demonstrate that ARP-T1 is a basal body protein and is involved in ciliogenesis by interacting with the ciliary machinery.” It should be toned down to something like this “Here, we show that ARP-T1 localizes to the basal body, interacts with several components of the ciliary machinery and contributes to cilium extension.”

Correction new line 75

Line 73-74: The authors overstate: “...give rise to the abnormally shortened cilia, and this may be caused by a disordered diffusion barrier.” ...as the integrity of the transition zone/diffusion barrier was not assessed in this study. It should be toned down to something like this “...give rise to the abnormally shortened cilia, and this may be caused by the displacement of septin 2.”

Correction new line 77-78

Line 77: Replace “We report that BDCS is the first ciliopathy of epidermal development and cancer.” by “We report for the first time the presence of cilia defects in ACTRT1-associated BDCS epidermal development and cancer, and propose that this pathology should be considered a ciliopathy.

Correction new line 81

Line 180: Replace “ARP-T1 associates with a ciliopathy in BDCS.” The authors should rephrase this title to: “ARP-T1-associated BDCS tissues have shorter cilia.”

Correction new line 171

Line 181- 182: “Based on the localization of ARP-T1 to the basal body and the implication of septin 2 in ciliary length control, we...”. The following clarification should help the reader to better understand the author's rationale for analyzing ciliary length: “Based on the localization of ARP-T1 to the basal body and its interaction with septin 2, that is involved in ciliary length control, we...”

Correction new line 172

Line 193-195: The authors should rephrase this sentence: “We concluded that BDCS, inherited BCC, unlike sporadic BCC, is a novel ciliopathy and the role of ARP-T1 as a part of cilia is to prevent the disease.”

to “These results suggest that ARP-T1 is important for cilia length and that ARP-T1-associated BDCS might originate from ciliary defects.

Correction new line 185-186

Line 220 – 221: “ARP-T1 favors for septin 2 localization in the axoneme, without affecting

the actin cellular network.” This sentence is not correct: the first half refers to the protein function while the second half refers to ARP-T1 loss/mutation/depletion. I suggest rephrasing to “ARP-T1 loss of function disrupts septin 2 localization to cilia without affecting the actin cellular network.”

Correction new line 207-208

Line 242: I suggest improving this title from “ARP-T1 is located in the midbody in dividing cells.” to “ARP-T1 also localizes to the midbody in dividing cells”.

Correction new line 233

Line 257-258: I suggest improving this sentence: “Both midbodies and PC contain acetylated tubulin, many proteins in the midbody can also be found at the base of the cilium in the centrioles (Fig. 5f-i).” to:

“Both midbodies and PC contain acetylated tubulin and several proteins at the midbody have been found to relocate to the base of assembling cilia to participate in ciliogenesis and in cilia-mediated signaling (Fig. 5f-i).”.

new line 248, we corrected: **“Both midbodies and primary cilia contain acetylated tubulin and several proteins, which have been found to relocate to the base of assembling cilia and to participate in ciliogenesis and in cilia-mediated signaling (Fig. 5f-i).”**

Reference “29” seems to be incorporated into the manuscript by mistake as it's completely unrelated to every sentence that it's associated with in the manuscript.

Indeed it was a mistake. We corrected the reference.

In lines 256 to 258 – References 28 to 30 are incorrectly used. Appropriate references must be used here. I assume that these mistakes were unintentional and I did not find any other cases of incorrect references but I strongly advise the authors to reconfirm all the references in their manuscript before submitting their final version!

We checked the references and corrected them if necessary.

Line 266 – Rephrase: “Here, we report that ARP-T1 is a ciliogenesis protein located at the basal body of PC.” To “Here, we report that ARP-T1 localizes to the basal body of PC where its involved in the regulation of the extension of the ciliary axoneme.”

Correction new line 254-255

Lines 327 to 338 need to be greatly clarified to make a more logical and easier to follow flow for the reader. If I understood the ideas in the text correctly then it might help to: -take the sentence “Despite all these clues,... in the shortened cilia observed in silenced ACTRT1 cells.” (line335-338)

-and move it to line327 after “...ciliogenesis.” and before “Similar to ARP-T1,...”

Correction new line 318-321

Line 327: “ciliary resorption” should be replaced by “shorter cilia or faster ciliary resorption”.

Correction new line 322

Line 339: Replace “Diffusion barrier...” by “The transition zone or diffusion barrier...”
Correction new line 331

Line 344-347: Replace to improve flow and avoid overstating: “Such a situation was different for septin 9, whose location remained unchanged, i.e. in the axoneme, septin filaments and the base of the PC. These results suggest that the ciliogenesis defect observed in ACTRT1 deficient cells could be due to a default in the diffusion barrier mediated by septin 2.” by:

“In contrast, the localization of septin 9 at the axoneme and base of cilia, and in septin filaments remained unchanged. These results suggest that the ciliogenesis defect observed in ACTRT1 deficient cells could be related to the mislocalization of septin 2, as it is required for normal axoneme extension (ref24).”

Correction new line 337-340

Line 347-349: “We are currently investigating the link between septin 2 and ARP-T1 in cilia formation, notably using SMO in/out assay and FRAP experiments. These results will be part of a forthcoming publication.” should be rephrased to a more standard form such as “Future experimental work will be needed to further investigate the link between septin 2 and ARP-T1 in cilia formation.”

Correction new line 340-342

Line 357: Replace “leads” by “contributes to” or “participates in”

We corrected the line 350: “...we propose a model where ARP-T1 **contributes to** primary ciliogenesis...”

Line 358: Replace “in” by “of the”

Correction new line 351

Line 388: I suggest replacing “or that another hit might be needed for a proper activation.” by “or that other elements might compensate to prevent a stronger activation of the pathway.”

Correction new line 381-382

Line 403: Replace “Our results show that ARP-T1 acts as a direct or indirect actor in a non-canonical HH pathway” by “Our results suggest that ARP-T1 is directly or indirectly involved in a non-canonical HH pathway”.

Correction new line 397

Line 405: Remove “...the centrosome, the centrioles and...” to simplify the extremely long sentence.

Correction new line 398

Line 406: Replace “Above all, our studies shed light on how ciliogenesis is controlled in carcinogenesis by ARP-T1” by “This study also sheds light on how ARP-T1-associated ciliary defects might contribute to carcinogenesis”...

Correction new line 399-401

“Our results suggest that ARP-T1 is directly or indirectly involved in a non-canonical HH pathway connecting the actin cytoskeleton organization involved in vesicle transport, basal body formation, and the formation of the primary cilium to prevent the pathogenesis of BDCS. This study also sheds light on how ARP-T1-associated ciliary defects might contribute to carcinogenesis.”